# A virtual burrow assay for head–fixed mice measures habituation, discrimination, exploration and avoidance without training

Andrew JP Fink[†*], Richard Axel, Carl E Schoonover[†*]

Department of Neuroscience, Howard Hughes Medical Institute, Columbia University, New York, United States

**Abstract** We have designed an assay that measures approach and avoidance behaviors in head-fixed mice at millisecond timescale, is compatible with standard electrophysiological and optical methods for measuring neuronal activity, and requires no training. The Virtual Burrow Assay simulates a scenario in which a mouse, poised at the threshold of its burrow, evaluates whether to exit the enclosure or to retreat inside. The assay provides a sensitive readout of habituation, discrimination and exploration, as well as avoidance of both conditioned and innately aversive cues.
DOI: https://doi.org/10.7554/eLife.45658.001

## Introduction

Mice in the wild must balance the need to seek safety inside their burrow with the need to gather information and resources in the outside world (*Birke et al., 1985*; *Blanchard and Blanchard, 2008*; *Blanchard and Blanchard, 1989*). An egress at the wrong moment, a delayed or slow ingress to escape from a predator, or even undue reticence to investigate the world outside all undermine the fitness of the organism. The behavioral motifs displayed at the threshold of the burrow and the circuits that govern them are therefore likely to have been selected over evolutionary time. We have developed a Virtual Burrow Assay (VBA) for head-fixed mice that aims to recapitulate the conditions characteristic of being poised at the threshold of the burrow in order to capture these innate behaviors.

Behavioral assays for head-fixed mice typically require several sessions of acclimation to head-fixation and task training, even for relatively simple applications such as measuring sensory discrimination (*Guo et al., 2014*) or obtaining controlled locomotion in a visual (*Dombeck et al., 2010*), or olfactory (*Radvansky and Dombeck, 2018*) virtual-reality environment. In training-dependent paradigms, variability in task proficiency can contaminate the measurement of the sensory or cognitive quantities that are inferred from the behavior. Moreover, training unavoidably engages circuits required for signaling reward, regulating satiety/motivation, learning the structure of the task, and learning, planning and executing new motor actions. This interposes additional layers of complexity between the sensory or cognitive operations that are the object of study and the behavior that is observed in order to infer their properties. This added complexity is likely to further challenge the already difficult interpretation of lesion or perturbation studies (*Wolff and Ölveczky, 2018*) or of the relationship between neural activity and concomitant behavior.

We developed the VBA in order to elicit interpretable behavior in head-fixed mice without requiring any training. The motivation for this was twofold: (1) to minimize behavioral noise introduced by variability in task proficiency and (2) to minimize the complexity of the circuit operations necessary to produce meaningful behavior under head fixation. When they are introduced into the VBA mice

**\*For correspondence:**
af2243@columbia.edu (AJPF);
ces2001@columbia.edu (CES)

[†]These authors contributed equally to this work

**Competing interests:** The authors declare that no competing interests exist.

exhibit an innate tendency to retreat inside an available enclosure—a drive that can be leveraged to measure a diversity of behaviors, ranging from innate avoidance and fear conditioning to stimulus discrimination and exploration. These behaviors, which are observed without training or prior acclimation to handling or head-fixation, are stereotyped and consistent across animals.

## Results

### The Virtual Burrow Assay and its operation

The VBA consists of a tube enclosure (virtual burrow), constrained to slide back and forth along the anterior-posterior axis of the body of a head-fixed mouse (*Figure 1A,B*). When placed in the virtual burrow mice retreat inside the enclosure, pulling the tube up around their bodies as far as possible—a behavior we have termed 'ingress' (*Figure 1—video 1*). All mice we have tested exhibit this behavior immediately after being introduced into the VBA, remaining fully inside the virtual burrow with only transient excursions from the ingress position (*Figure 1C*). Neither training nor acclimation to the assay is required to observe this consistent and sustained ingress.

The assay measures the propensity of an animal to ingress in response to a given stimulus. It is therefore necessary to pull the virtual burrow away from the mouse in order to ask what induces the animal to ingress. When the enclosure is retracted by a motor tethered to the virtual burrow, mice invariably try to get back inside (*Figure 1D*, *Figure 1—video 2*). After an initial bout of resistance (<30 s), mice voluntarily maintain an 'egress' position and thus can be presented test stimuli. Mice quickly adapt to this scenario, exhibiting within one to three minutes periods of >15 s during which they voluntarily hold the virtual burrow in the egress position. This sequence of behaviors has been observed in all mice tested and been found to persist across repeated tests on four separate days over a 16 day interval (*Figure 1—figure supplement 1*). No prior acclimation to handling or head fixation is required.

### Air puffs elicit rapid and reliable ingress

We first determined whether an innately aversive stimulus, such as air puff, induces ingress, and if so, the degree to which this response is rapid and stereotyped (*Figure 2A*). Strong air puffs delivered to the snout (80 psi, 2 mm distance) elicited short-latency, rapid ingress in all mice tested on all trials (*Figure 2B*; *Figure 2—video 1*). Animals generated this behavior by pulling the burrow up to the ingress position in a coordinated, simultaneous movement of their fore- and hind-limbs. The latency of ingress varied little across animals ($\mu$ = 18.5 msec, $c_v$ = 0.07, N = 5 mice, three trials per mouse) and across trials ($c_v$ = 0.10). Weak air puffs (2 psi, 15 cm distance) did not elicit ingress but instead a transient movement resulting in no net change in burrow position. This apparent flinch-like startle response (*Davis, 1984*) can be clearly distinguished from the ongoing movement of the burrow caused by the animal's breathing (*Figure 2C*).

We then demonstrated that ingress in response to air puff reflects flight to shelter inside the virtual burrow, rather than just a backwards movement to avoid the source of the stimulus. First, we compared responses to air puffs delivered to the snout with air puffs delivered to the hindquarters (*Figure 2D* top, orange vs. green). As observed previously, all animals ingressed on all trials when air puffs were directed at the snout (*Figure 2E* orange and 2F left, N = 4 mice, three trials each). When air puffs were directed at the hindquarters, mice also invariably ingressed (*Figure 2E* green and 2F second from left, N = 4 mice, three trials each). Thus the drive to ingress into the virtual burrow overcomes the drive to move in the direction opposite the air puff. Second, we replaced the tube with a flat, open platform that was similarly constrained to move along the anterior-posterior axis of the mouse's body (*Figure 2D* bottom). Mice in this configuration exhibited a transient, flinch-like response to air puff delivered to the snout or hindquarters (N = 4 mice each, three trials each) followed, in some cases, by relatively small displacements of the platform with little net preference between the egress or ingress directions (*Figure 2E* pink and blue, 2F second from right and right, median displacement of 14.9 mm and 17.0 mm in tube vs. 0.2 mm and 2.3 mm on platform, for snout and hindquarters, respectively). These observations, together with the stereotypy, high reproducibility across animals, and low latency of the response to strong air puff, and the fact that these behaviors require no training, suggest that ingress to air puff in the VBA reflects an innate behavioral program to flee to shelter.

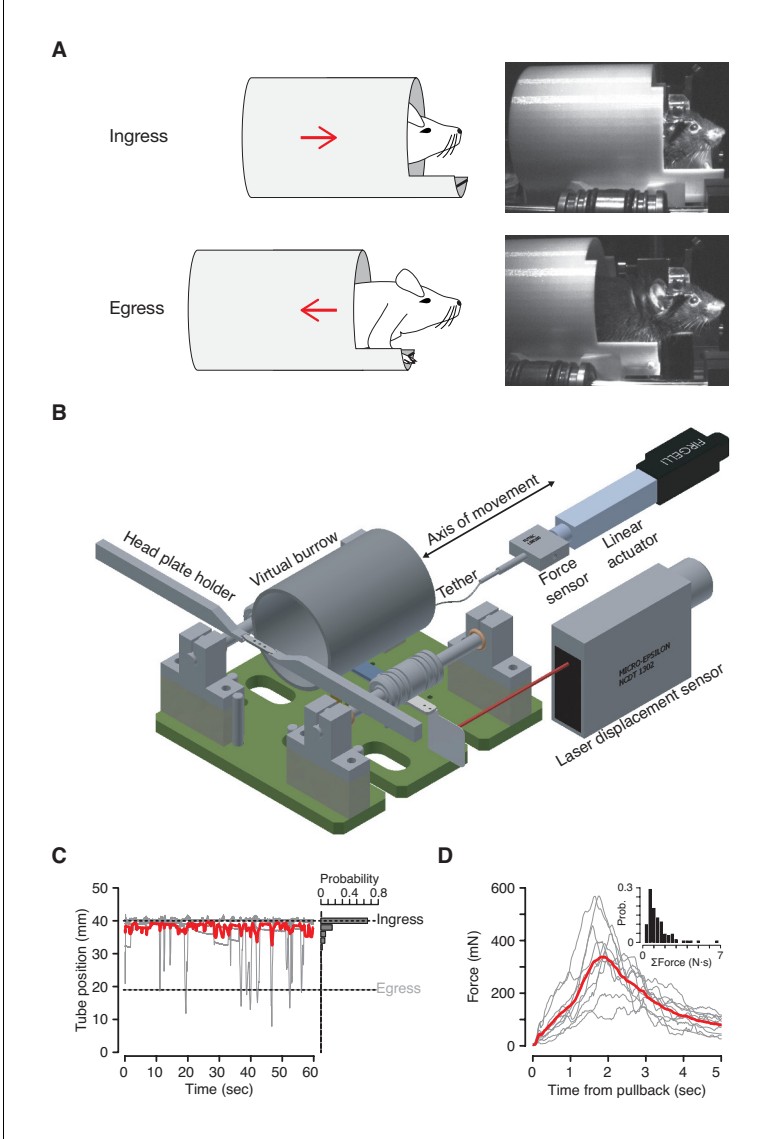

**Figure 1.** The Virtual Burrow Assay. (**A**) Left; diagram of ingress (top) and egress (bottom). Right, photograph of mouse in ingress position (top) and egress position (bottom). (**B**) Instrument diagram. The mouse's head is fixed by a headplate holder while it stands inside a virtual burrow. The burrow's movement is constrained to the animal's anterior-posterior axis by a pair of near-frictionless air bearings. A linear actuator can retract the burrow, pulling the animal out of the enclosure. A laser displacement sensor measures burrow position and a force sensor measures any resistance the animal exerts against the tether. (Rendering courtesy of T. Tabachnik, Advanced Instrumentation, Zuckerman Mind Brain Behavior Institute). (**C**) Tube position during 60 s of open loop mode. Gray, individual animals; red, mean across N = 10 mice. For visual clarity only five randomly selected individual (gray) traces are shown. Dashed lines indicate ingress and egress positions. Histogram at right shows distribution of tube position during the same epoch across all animals. (**D**) Force (in millinewtons) generated during first 5 s after pullback to egress position for N = 10 mice (gray traces, mean force generated across all pull-backs for each individual animal; red trace, mean across all animals). Inset: distribution of total force exerted each time the animals were pulled back. Force was integrated after each pullback until the animal ceased resisting and the tether was slackened.

DOI: https://doi.org/10.7554/eLife.45658.002

The following video and figure supplements are available for figure 1:

**Figure supplement 1.** Persistence of stereotyped behavior across multiple days.

DOI: https://doi.org/10.7554/eLife.45658.004

**Figure supplement 2.** Schematic and flow diagram.

*Figure 1 continued on next page*

*Figure 1 continued*

DOI: https://doi.org/10.7554/eLife.45658.003

**Figure 1—video 1.** Spontaneous behavior of a mouse in the Virtual Burrow Assay.

DOI: https://doi.org/10.7554/eLife.45658.005

**Figure 1—video 2.** Burrow pulled to egress position.

DOI: https://doi.org/10.7554/eLife.45658.006

## Looming visual stimuli elicit ingress

We next asked whether visual stimuli that evoke flight in freely moving animals also evoke a response in the VBA. For this we employed a predator-like visual 'looming' stimulus known to trigger flight in freely moving mice (*De Franceschi et al., 2016*; *Yilmaz and Meister, 2013*) (*Figure 3A*, top left). Mice in the VBA ingressed on 80% of trials in response to an expanding black disk displayed above their heads (*Figure 3A*, top left, B, top, C, orange, and D, left; *Figure 3—video 1*). The habituation to repeated stimulation reported in freely moving mice (*De Franceschi et al., 2016*) was not observed in the VBA (*Figure 3B*, top). We also presented stimuli to mice in the VBA that do not elicit flight responses in freely moving mice: a contracting black disk and a small black disk sweeping across the visual field (*Figure 3A*, top middle and top right) (*De Franceschi et al., 2016*). These stimuli did not reliably elicit ingress, yielding responses on 0% or 13% of trials, respectively (*Figure 3B*, middle and bottom, C, pink and green, D, middle and right). These data show that a visual stimulus that elicits flight in freely moving mice elicits ingress in the VBA and that control stimuli that do not elicit flight do not elicit ingress.

## Unanticipated odorant stimuli elicit ingress or exploration

We next asked whether the VBA can measure habituation (*Figure 4*). We observed that mice reliably ingress to initial presentations of an odorant stimulus, and their response decreases over the course of repeated stimulation (ingress on 67% of first three trials; 10% of subsequent trials, N = 5 mice, *Figure 4B*, left, *Figure 4C*, orange, and *Figure 4D*, left). This response is odor-selective: after 15 presentations of the first odorant (Odor 1) we randomly interleaved it with a second odorant (Odor 2) and observed that animals selectively ingressed to the second, but remained habituated to the first odor (ingress on 47% of first three Odor 2 trials; 12% of subsequent trials, *Figure 4B,C* blocks 16–30, and *Figure 4D*, middle). After 15 further blocks of Odor 1 and Odor 2, resulting in habituation to the second odor, we randomly interleaved them with a third odorant (Odor 3), and once again observed selective ingress to the new odor (ingress on 60% of first three Odor 3 trials; 8% of subsequent trials, *Figure 4B,C* blocks 31–45, and *Figure 4D*, right).

The selective response to a second odor following habituation to the first (*Figure 4B–D*) demonstrates that the response decrement is due neither to adaptation of the sensory epithelium nor to effector fatigue (*Groves and Thompson, 1970*; *Rankin et al., 2009*; *Thompson and Spencer, 1966*). A habituated response can transiently increase following presentation of a different stimulus, a phenomenon termed dishabituation (*Groves and Thompson, 1970*; *Thompson and Spencer, 1966*). However, we did not observe renewed ingress to Odor 1 following the first presentation of Odor 2 or Odor 3 (See Materials and methods section for discussion).

On early trials mice occasionally exit the virtual burrow before initiating their ingress (data not shown). We speculated that this voluntary egress corresponds to a brief bout of exploration. To test this, we coupled the odor source to the burrow, so that egress from the burrow brought the odor source closer to the animal's nose, while ingress distanced it (*Figure 4E*). We reasoned that granting the animal control over its proximity to the odor port would allow it to select between the drive to explore an odorant stimulus (by exiting the burrow) and the drive to remain inside. In contrast to all other experiments, in which trials were initiated while the animal was in a mandatory egress position, here the animals were granted control over burrow position at all times and almost invariably maintained full ingress prior to stimulus delivery. We observed that under this configuration mice exited the virtual burrow for brief (~1 s) bouts before resuming a fully ingressed position (*Figure 4F*, *Figure 4—video 1*). This response habituated, with the animals egressing further on earlier than on later trials (median egress 21.9 mm for trials 1–2 vs. 10.5 mm for trials 3–5, N = 5 mice, *Figure 4G,H*). Together, these results indicate that the VBA measures selective responses to unanticipated

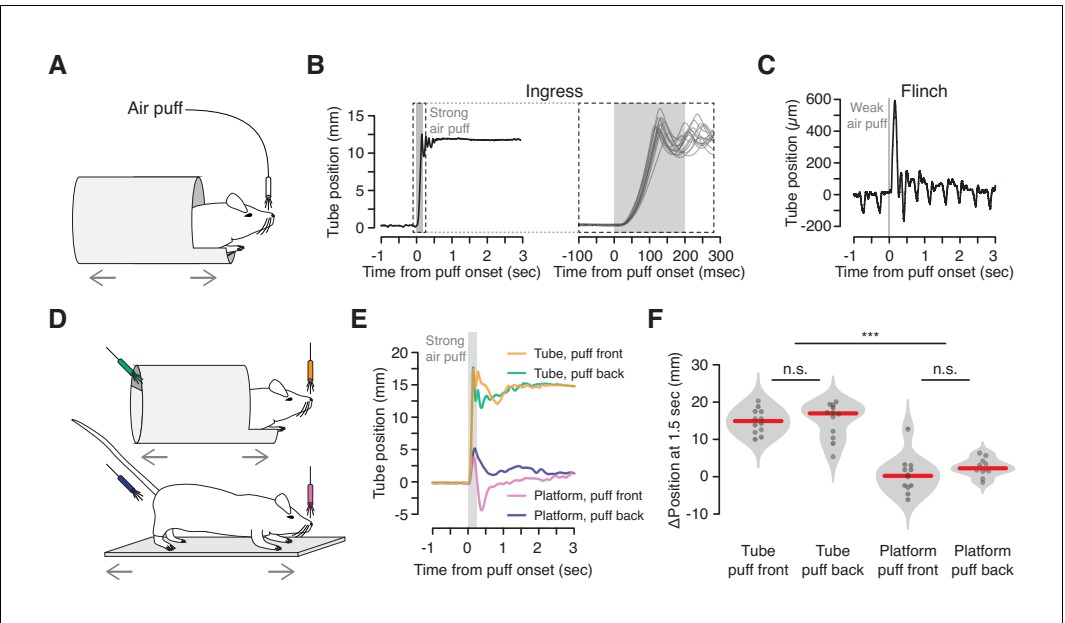

**Figure 2.** Reliable, short-latency ingress to noxious air puff. (**A**) Diagram of experimental set up. The mouse is head-fixed in the virtual burrow and an air puff is delivered to the nose. (**B**) Left, burrow position over time showing a single ingress in response to a strong air puff (gray box, 200 msec, 80 psi). Upward deflections correspond to burrow movement towards the animal's head (ingress). Right, 15 ingress responses from a single animal to 15 air puffs. Dashed box at left demarcates epoch in which time scale is expanded at right. (**C**) Example of flinch in response to weak air puff. Downward going, approximately 2 Hz oscillations correspond to the animal's breathing cycle. Upward going low-amplitude, transient deflection corresponds to startle in response to air puff (gray box, 20 msec, 2 psi). (**D**) Diagram of tube (top) and platform (bottom) variants. Air puffs are directed either at the snout (top, orange and bottom, pink) or at the hindquarters (top, green and bottom, blue). (**E**) Mean change in burrow position in response to air puff (gray box) across all animals and all trials (N = 4 animals, three trials each, per condition); same color scheme as in previous panel. (**F**) Change in burrow position at T = 1.5 s relative to pre-stimulus epoch, pooled across animals and trials. A Wilcoxon rank-sum test was employed to evaluate whether the change in burrow position differed significantly (p(tube-front,tube-back)=0.71, p(tube-front,platform-front)=$9.7\times10^{-05}$, p(tube-front,platform-back)=$3.7\times10^{-05}$, p(tube-back,platform-front)=$9.7\times10^{-05}$, p(tube-back,platform-back)=$6.0\times10^{-05}$, p(platform-front,platform-back)=0.078, N = 4 mice, three trials each, per condition). Individual trials, gray points. Normalized, smoothed histogram, light gray shading. Median, red line. *** indicates p<0.001, n.s. indicates p≥0.05.

DOI: https://doi.org/10.7554/eLife.45658.007

The following video is available for figure 2:

**Figure 2—video 1.** Ingress in response to strong air puff.

DOI: https://doi.org/10.7554/eLife.45658.008

stimuli—ingress or egress, depending upon the configuration of the assay—and that these behaviors can be employed to measure habituation.

## Aversively conditioned odorants elicit ingress

We next asked whether the VBA can measure conditioned responses following aversive Pavlovian conditioning (*Figure 5*). One odorant (CS+) was paired with footshock in a conditioning chamber; a second (CS-) was presented in the chamber without footshock; and a third (Odor3) was not presented in the chamber but was tested in the VBA before and after conditioning.

Prior to conditioning, mice initially ingressed in response to all three stimuli then habituated over subsequent trials (*Figure 5D,F,H*). After conditioning (48 hr later) the response recovered (*Figure 5C,G*, first trial), and the animals once again habituated to the CS- and Odor3. However, they continued to ingress to the CS+, exhibiting both a greater ingress likelihood (CS+ 72%, CS- 26%, Odor3 33%, mean across trials 2–7, six trials per mouse, pooled across N = 9 mice) as well as a

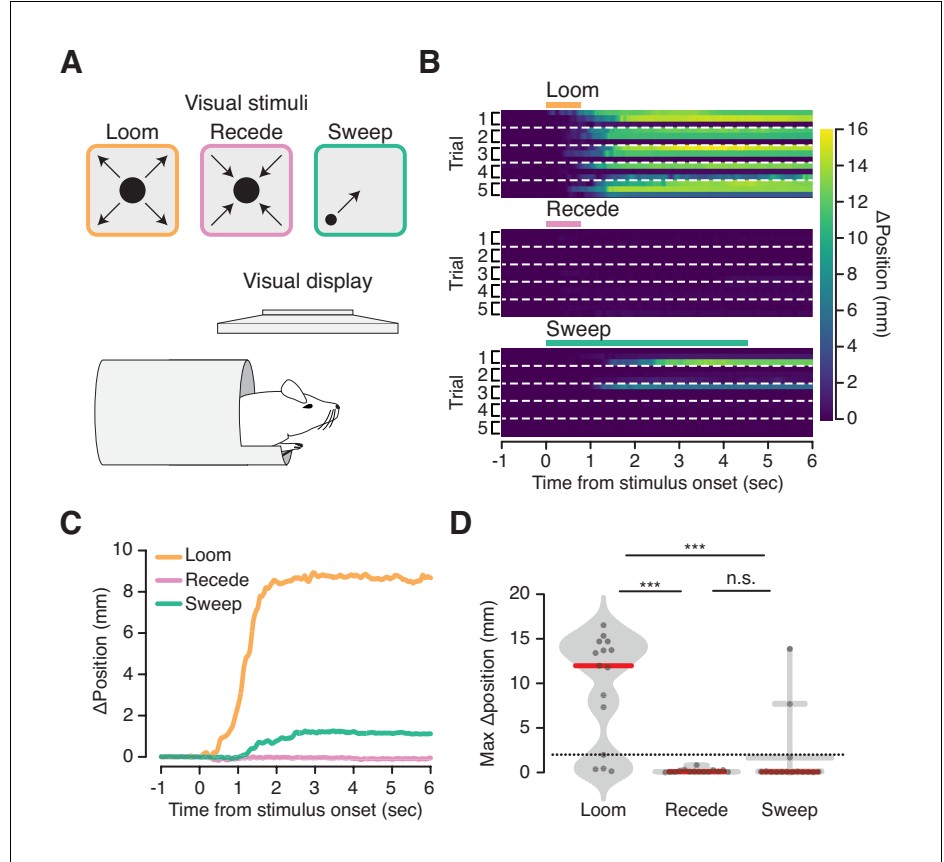

**Figure 3.** Flight-inducing visual stimuli selectively elicit ingress. (**A**) An expanding black disk (left, orange), a contracting black disk (middle, pink), and a sweeping black disk of constant size (right, green), were presented on a visual display positioned directly over a mouse head-fixed in the virtual burrow (bottom). (**B**) Responses of all mice on all trials to three visual stimuli (three mice per condition, five trials each) ordered by mouse within each trial: Expanding ('Loom'), disk widening from 2° to 50° over 250 msec, holding the 50° disk for 500 msec; Contracting ('Recede'), disk diminishing from 50° to 2° over 250 msec, holding the 2° disk for 500 msec; Sweeping ('Sweep'), 5° disk sweeping smoothly across the diagonal of the screen at a rate of 21°/sec. Color map corresponds to change in burrow position with respect to baseline. Dashed lines separate trials. (**C**) Mean change in burrow position per condition across all animals and all trials. (**D**) Maximum change in burrow position in the 6 s following stimulus onset per condition across all animals and all trials. Ingress was defined as a maximum displacement of the burrow relative to the pre-stimulus baseline position >2 mm, indicated by the dashed line. The likelihood of ingress was 0.73, 0.00 and 0.13 for loom, recede, and sweep, respectively. Individual trials, gray points. Normalized, smoothed histogram, light gray shading. Median, red line. A two-proportion z-test pooled across all mice (N = 3) was employed to evaluate whether the probability of ingress differs significantly across stimulus conditions. p(loom,recede)=$1.5 \times 10^{-05}$, p(loom,sweep)=$4.6 \times 10^{-04}$, p(recede,sweep)=0.072. *** indicates p<0.001, n.s. indicates p≥0.05.

DOI: https://doi.org/10.7554/eLife.45658.009

The following video is available for figure 3:

**Figure 3—video 1.** Ingress in response to visual looming stimulus.

DOI: https://doi.org/10.7554/eLife.45658.010

---

greater displacement (median ingress 9.98 mm for CS+, 0.27 mm for CS-, and 0.36 mm for Odor3, N = 9 mice, *Figure 5B,C,E,G,I*) in response to the CS+ than to the CS- or Odor3.

On occasion we noted a second behavioral response evoked by the CS+ following conditioning: an oscillation in burrow position (*Figure 5—figure supplement 1*). Simultaneous video recording (not shown) indicated that this high frequency oscillatory response was associated with trembling of the animal's body. While we observed ingress without trembling, on some trials trembling preceded ingress by several seconds (*Figure 5—figure supplement 1*, top trace), or occurred on ingress-free

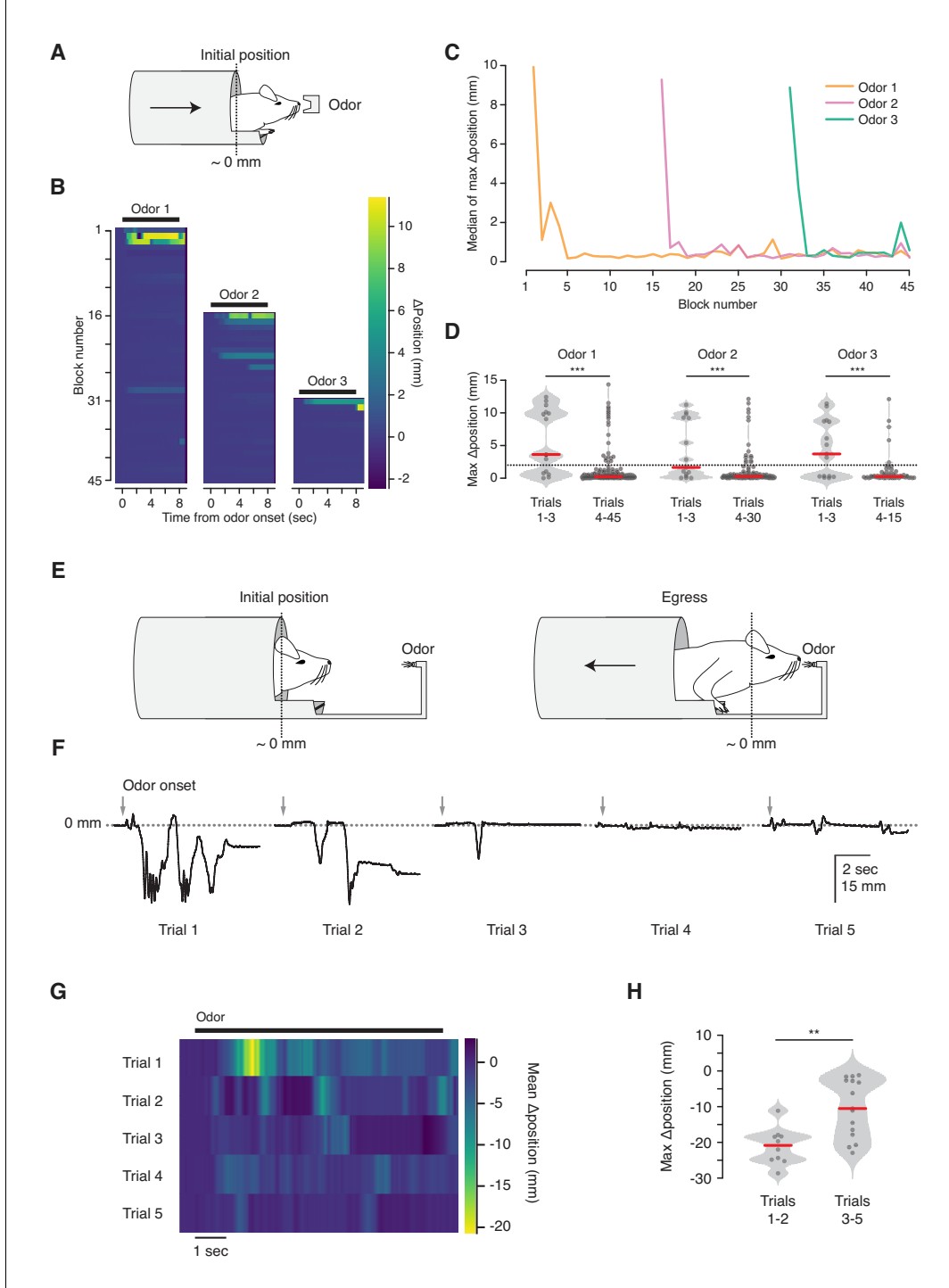

**Figure 4.** Habituation to unanticipated stimuli. (**A**) Three odorant stimuli were delivered to mice in the virtual burrow assay. (**B**) Habituating responses to repeated presentation of odorant stimuli from a representative mouse. Color map corresponds to change in burrow position with respect to baseline. Black lines indicate odorant stimulus epoch (8 s duration). (**C**) Median value across mice (N = 5) of maximum change in burrow position, per odor condition, per block. (**D**) Maximum change in burrow position for each odorant during the first three trials (left) and all later trials (right), pooled across animals. Individual trials, gray points. Normalized, smoothed histogram, light gray shading. Median, red line. A two-proportion z-test on ingress probability pooled across all mice (N = 5) was employed to evaluate whether the probability of ingress differed significantly between the first three presentations of each odorant stimulus and all subsequent presentations of that odorant; ingress defined as

*Figure 4 continued on next page*

*Figure 4 continued*
maximum displacement >2 mm during the 8 s stimulus epoch, indicated by the dashed line. p(Odor1 trials 1–3, Odor1 trials 4–45)=$8.8 \times 10^{-10}$, p(Odor2 trials 1–3, Odor2 trials 4–30)=$1.9 \times 10^{-04}$, p(Odor3 trials 1–3, Odor3 trials 4–15)=$2.2 \times 10^{-06}$. *** indicates p<0.001. (E) Diagram of the odor port coupled to the virtual burrow; the mouse is required to egress in order to draw the odor source closer to its nose. (F) Habituating response of a representative mouse to repeated presentations of an odorant stimulus. Downward-going traces correspond to egress. Gray arrows indicate odorant stimulus onset (8 s duration). (G) Average response of all mice tested (N = 5) per trial. Color map as above, except that warmer colors depict egress rather than ingress. Note that, as above, 0 mm corresponds to the virtual burrow's position prior to stimulus presentation; however in these experiments the animals began each trial in the ingress position rather than in the egress position. (H) Maximum change in burrow position during the first two trials (left) and the subsequent three trials (right), pooled across animals. A Wilcoxon rank-sum test was employed to evaluate whether the maximum change in burrow position differed significantly (p(trials1-2,trials3-5)=0.0021, N = 5 mice). Individual trials, gray points. Normalized, smoothed histogram, light gray shading. Median, red line. ** indicates p<0.01.
DOI: https://doi.org/10.7554/eLife.45658.011
The following video is available for figure 4:
**Figure 4—video 1.** Egress in response to unanticipated odorant stimulus.
DOI: https://doi.org/10.7554/eLife.45658.012

trials (*Figure 5—figure supplement 1*, middle trace), and was selective for the CS+ following conditioning. Thus the VBA can measure two kinds of stimulus-selective conditioned responses following aversive conditioning: trembling and ingress.

We took advantage of the high temporal precision of the VBA to measure the exact latency of ingress responses to CS+ presentations. Mice can report discrimination of odor stimuli within ~100 msec (*Resulaj and Rinberg, 2015*) and we have demonstrated that they are capable of initiating stimulus-evoked ingress within ~20 msec (*Figure 2*). It was therefore surprising to observe that the median latency to ingress to the CS+ was 709 msec, and 27% of conditioned respones (CRs) were initiated at latencies higher than 1 s following CS+ onset (*Figure 6A,B*). Precise measurement of CR onset revealed a second unanticipated feature: the frequency of high-latency (>1 s) responses increased significantly as a function of trial number, from 4.2% on the first three trials to 39% on last three trials (*Figure 6C*). Therefore, extinction is characterized not only by a decrease in the probability of a CR but also by an increase in the latency to produce a CR.

## Neurophysiological recording in the VBA

We next tested whether it is possible to obtain stable extracellular recordings of single units in the brains of mice while they are behaving in the VBA. For this we implanted a silicon probe in layer 2 of the piriform cortex and recorded spontaneous spiking activity after head-fixation in the apparatus. No olfactory stimuli were presented during this experiment. After spike sorting we compared both mean and single-unit firing rates to three behavioral quantities: position, absolute velocity, and force. During abrupt transitions in behavioral state, neither mean nor single-unit rates appeared to be affected (*Figure 7A*). Indeed, the variance in mean firing rate accounted for by position, velocity and force is negligible (*Figure 7B*, top; linear regression $R^2 = 9.3 \times 10^{-3}$, $8.9 \times 10^{-3}$, and $3.4 \times 10^{-3}$, respectively). This observation also holds at the single-unit level (*Figure 7B*, bottom; linear regression median $R^2 = 3.0 \times 10^{-4}$, $1.8 \times 10^{-3}$, and $1.1 \times 10^{-3}$, respectively). We conclude that behavior in the VBA does not sufficiently disrupt simultaneous neurophysiological recording as to prevent or cause spurious detection of spike waveforms. The assay also permits stable measurement of calcium transients in response to odorant stimuli using 2-photon microscopy through a GRIN lens implanted in the basolateral amygdala (O'Neill, P.K., personal communication).

## Discussion

We have designed an assay for head-fixed mice that reveals a diverse set of behavioral features related to sensory, cognitive, emotional and motor functions. These features can be measured without the animal having been trained in the sensorimotor contingencies of the instrument, or even been previously acclimated to head-fixation. The assay captures flight-like behavior (*Figures 2* and *3*), exploration (*Figure 4*), as well as the detection of the precise onset of conditioned responses to

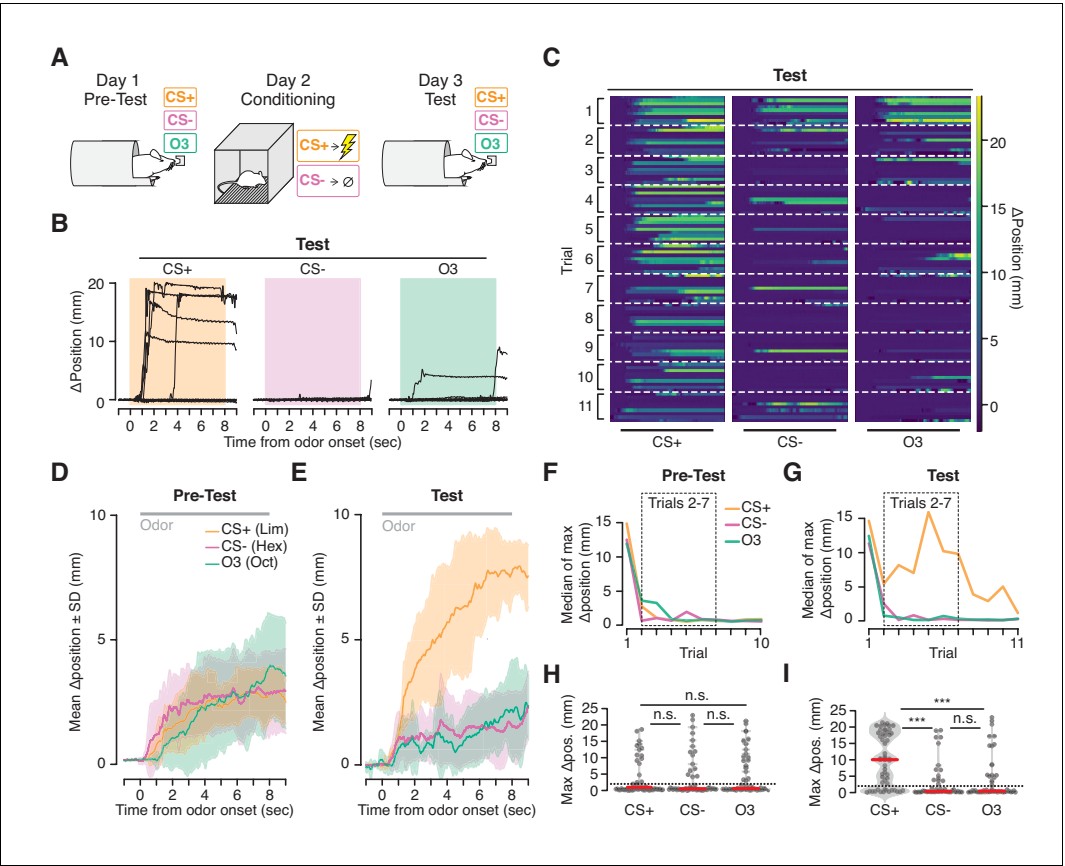

**Figure 5.** Ingress in response to aversively conditioned odorant stimuli. (**A**) Three odorant stimuli were presented to mice head fixed in the VBA on day 1 (Pre-test) and day 3 (Test). On day 2 (Conditioning), animals were placed in a fear conditioning chamber and two of the odorant stimuli were presented: a CS +odorant, paired with shock, and a CS- odorant, never paired with shock. A third odorant Odor3 (O3) was presented on days 1 and 3 but not during conditioning on day 2. (**B**) Change in burrow position relative to pre-stimulus baseline on individual trials after odor-shock conditioning from a representative mouse. Colored box demarcates odorant stimulus epoch. CS +: paired with shock; CS-: presented in chamber without shock; Odor3: not presented in fear conditioning chamber. (**C**) Test responses of all mice on all trials to the three stimuli, ordered by mouse within each trial. Lines at bottom indicates odorant stimulus epoch (8 s duration). D, E. Mean change in burrow position during Pre-test (**D**) and Test (**E**) relative to pre-stimulus baseline per odor condition during trials 2–7 (shading indicates ±1 standard deviation, N = 9 mice). Gray line at top corresponds to odorant stimulus epoch. F, G. Median value across mice of maximum change in burrow position, per odorant condition, per trial during Pre-test (**F**) and Test (**G**). Dashed boxes demarcate trials 2–7, used to compute mean responses in D and E, and to perform statistical tests in H and I. H, I. Maximum change in burrow position during the odorant stimulus, per condition across all animals on trials 2 through seven during Pre-test (**H**) and Test (**I**). Individual trials, gray points. Normalized, smoothed histogram, light gray shading. Median, red line. The probability of ingress for each odorant stimulus during Test was 0.72 for CS+, 0.26 for CS-, and 0.33 for Odor3. A two-proportion z-test on ingress probability on trials 2–7 (6 trials per mouse, pooled across N = 9 mice) was employed to evaluate whether the probability of ingress differed significantly; statistical analysis was restricted to trials 2–7 to mitigate the effects of recovery observed on the first trial and extinction observed on the last four trials; ingress defined as maximum displacement >2 mm during the 8 s stimulus. p(CS+,CS-)=$1.0\times10^{-05}$, p(CS+,O3)=$5.0\times10^{-05}$, p(CS-,O3)=0.35. Ingress threshold indicated by the dashed line; *** indicates p<0.001, n.s. indicates p>0.05. This result is robust to the choice of ingress threshold over a range of 0.5 to 10 mm (**Figure 5—figure supplement 2**).

DOI: https://doi.org/10.7554/eLife.45658.013

The following figure supplements are available for figure 5:

**Figure supplement 1.** Trembling in response to aversively conditioned odorant stimuli.
DOI: https://doi.org/10.7554/eLife.45658.014

**Figure supplement 2.** Robustness of statistical test to choice of ingress threshold.
DOI: https://doi.org/10.7554/eLife.45658.015

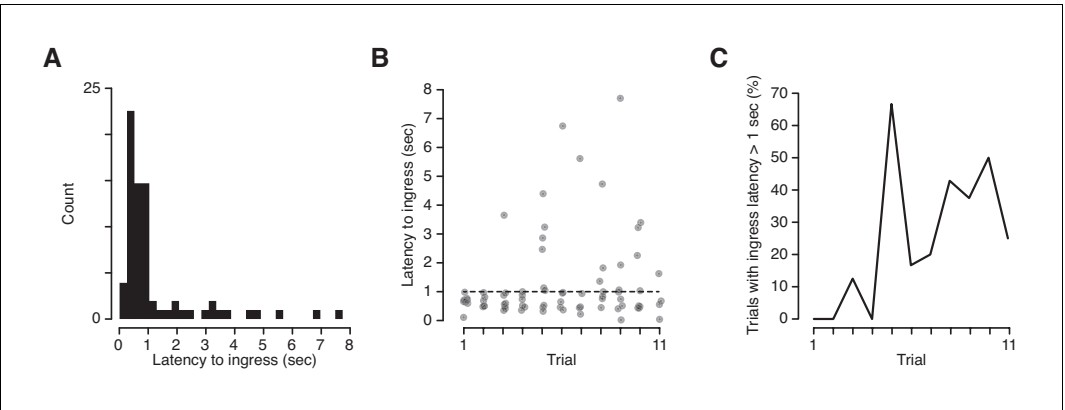

**Figure 6.** Latency of ingress to aversively conditioned odorant stimuli. (**A**) Distribution of ingress onset latency during CS+ trials for all mice. Ingress onset defined as the first sample in which displacement exceeded 0.75 mm following stimulus onset. (**B**) Ingress onset latency for all mice as a function of trial number. Dashed line indicates threshold for high latency ingresses (1 s). (**C**) Percentage of ingresses whose latency exceeded 1 s. The Spearman's rank correlation coefficient was computed to measure the strength of the relationship between the fraction of high-latency ingress and trial number ($r_s$ = 0.72, p=0.013).
DOI: https://doi.org/10.7554/eLife.45658.016

aversively conditioned odor stimuli (*Figures 5* and *6*)—measurements that have otherwise not been possible in head-fixed mice. In contrast to standard training-based assays for head-fixed mice, this training-free approach may reduce variability due to differences in task proficiency across animals and minimize the complexity of the circuit operations necessary to produce meaningful behavior under head fixation. The apparatus is compatible with standard electrophysiological and optical methods for measuring neuronal activity (*Figure 7*). Because it measures behavior at a timescale comparable to that of neuronal dynamics, the assay permits direct comparison between these two quantities.

Four observations suggest that ingress in the VBA reflects an innate behavioral program to seek the safety of an enclosure. (1) Mice exhibit consistent and reliable ingress without training or even acclimation to head-fixation (*Figure 1C*) and invariably resist being pulled out of the burrow (*Figure 1D*). (2) Ingress responses to air puff are characterized by markedly low latency and low variability across trials and across animals (*Figure 2A–C*), indicative of a highly conserved behavior. (3) Ingress does not depend upon the location of the source of the air puff but does require the availability of an enclosure (*Figure 2D–F*). (4) Mice ingress selectively to looming stimuli (*Figure 3*) that evoke flight in freely moving mice (*De Franceschi et al., 2016*; *Yilmaz and Meister, 2013*). We speculate that by simulating key features of the mouse *Umwelt* (*von Uexküll, 1957*), the contingencies of the assay permit expression of these innate behavioral programs in spite of the contrivance of the apparatus and of head-fixation.

We have observed four stimulus-induced behaviors in this assay: flinch, ingress, egress bout, and trembling. We interpret flinch in response to mild air puff as a startle response (*Davis, 1984*); ingress in response to puff and looming stimuli as a flight-like response; egress bout as exploration; and trembling as a component of what is commonly referred to as 'freezing'. Indeed, while mice typically ingressed in response to CS+ presentation (*Figure 5*), on some trials animals also exhibited a selective trembling response to the aversive cue (*Figure 5—figure supplement 1*). We made this same observation in high-speed video taken of freely moving animals during training in the conditioning chamber (data not shown), confirming previous reports of high-frequency trembling in wild rodents following the presentation fear-inducing stimuli (*Griffith, 1920*; *Hofer, 1970*). We also observed two distinct responses to unanticipated stimuli, which depend on the context in which the stimulus was presented. When unanticipated stimuli were presented to mice in the exposed, egress position, they reliably ingressed (*Figure 4A–D*). In contrast, mice briefly egressed in order to sample unanticipated stimuli if these were presented while the animals were ingressed (*Figure 4E–H*). This parallels the behavior of unconstrained animals, which exhibit either exploration (neophilia), avoidance

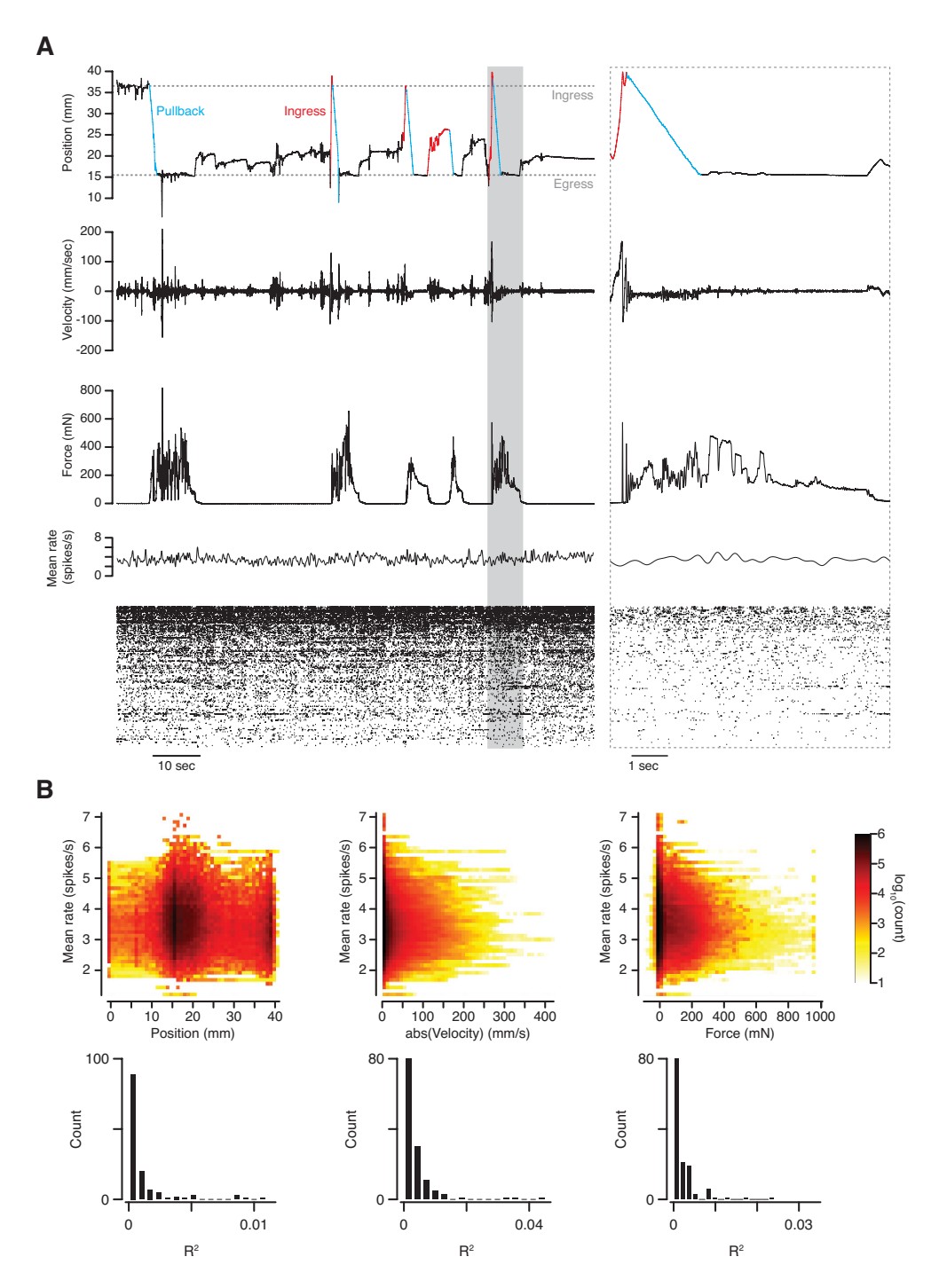

**Figure 7.** Simultaneous measurement of behavior and neuronal activity. (**A**) Top to bottom: position, velocity, force, mean firing rate and raster plot of 133 simultaneously recorded single-units in anterior piriform cortex of an awake mouse in the VBA. Left; 2 min epoch that includes the first pullback following transition to closed-loop mode at approximately T = 10 s; grayed-out box at left demarcates epoch in which time scale is expanded at right; blue portions of position trace correspond to pullbacks, red portions to spontaneous ingresses that exceeded $\theta_p$, resulting in pullback. (**B**) Top: three-dimensional histograms of mean firing rate vs. position (left), absolute velocity (middle) and force (right) during an 89-mn recording. Bottom, $R^2$ of linear regressions of firing rate on position, absolute velocity, and force for individual single-units.
DOI: https://doi.org/10.7554/eLife.45658.017

(neophobia), or both in sequence, in a manner that depends on context (*Berlyne, 1950*; *Gershman and Niv, 2015*).

Many standard assays of rodent behavior in the laboratory measure events that unfold over seconds or minutes. These include time spent freezing (*Bouton and Bolles, 1980*; *Griffith, 1920*), conditioned suppression (*Estes and Skinner, 1941*), habituation (*Groves and Thompson, 1970*; *Thompson and Spencer, 1966*), approach versus avoidance (*Young, 1959*), conditioned place preference (*Garcia et al., 1957*), and exploration (*Ennaceur and Delacour, 1988*). The VBA measures the onset of ingress and egress with millisecond precision, thus permitting fine alignment of behavior with neural dynamics. Such alignment to sharp transitions in behavioral state has proven fruitful in primate neurophysiology: for instance, alignment to the precise time of eye saccades indicating a perceptual decision permits the investigation of the neuronal events underlying a decision process (*Roitman and Shadlen, 2002*).

The VBA can exploit stimulus-selective habituation to implement a discrimination assay without training. This strategy has been employed across many experimental systems, ranging from the olfactory system of freely moving rodents (*Cleland et al., 2002*) to the visual system of human infants (*Friedman, 1972*), and can be exploited to construct psychometric curves for detection or discrimination. Many head-fixed assays are capable of measuring detection and discrimination (e.g. *Guo et al., 2014*), but these typically depend upon instrumental conditioning, requiring acclimation to head-fixation and training in the sensorimotor contingencies of the task; moreover, these are likely to result in perceptual learning (*Fahle and Poggio, 2002*), causing overestimation of default perceptual performance.

In *Figure 5*, we demonstrate that the VBA can be employed to measure fear conditioning. Previously-described paradigms (such as *Lovett-Barron et al., 2014*) have successfully translated conditioned lick-suppression (*Bouton and Bolles, 1980*) to a head-fixed preparation, but this measurement is characterized by relatively low temporal resolution both in freely moving and in head-fixed mice. In *Figure 6* we leverage the VBA's fine temporal resolution to measure the precise latency of ingress in response to CS+ presentation. We observed a median latency of 709 msec, with over one quarter of responses initiating after longer than 1 s. These latencies cannot be explained in terms of delays in sensory detection (*Resulaj and Rinberg, 2015*) or ingress motor pattern generation (*Figure 2*). A circuit centered around the amygdala has been proposed to orchestrate Pavlovian learning and the resultant expression of conditioned responses (*Johansen et al., 2011*), but these results suggest that the expression of Pavlovian fear conditioning engages additional processing not accounted for in this model. Furthermore, it does not explain how the probability of long (>1 s) latencies might increase over the course of extinction.

The wide range of responses measurable by the VBA yields sensitive indicators of sensory detection and discrimination thresholds, exploration, neophobia, memory, and motor function—behavioral features that may be affected in models of psychiatric and neurological diseases. For example, in preliminary experiments we have found that animals habituate to odorant stimuli more rapidly when administered anxiolytics; and moreover, we have observed that the rate at which chronically stressed mice acclimate to the closed loop contingencies is diminished by the administration of selective serotonin reuptake inhibitors. Moreover, key features of motor function—including balance, tremor, chorea, limb and core strength, movement initiation, and movement velocity—can be precisely measured by tracking the position of the burrow and the force exerted against the tether.

## Materials and methods

### Subjects and surgery

All procedures were approved by the Columbia University Institutional Animal Care and Use Committee (protocols AC-AAAI8650 and AC-AAAT5466). 8–17 week old, male C57BL/6J mice (Jackson laboratories, Bar Harbor, ME) were fitted with a titanium head plate (27.4 mm x 9.0 mm x 0.8 mm, G. Johnson, Columbia University). Animals were anesthetized with isoflurane (3% induction, 1.5–2% maintenance) and placed within a stereotaxic frame (David Kopf Instruments, Tujunga, CA) on a feedback-controlled heating pad (Fine Science Tools, Foster City, CA). Carprofen (5 mg/kg) was administered via subcutaneous injection as a preoperative analgesic and bupivacaine (2 mg/kg) was delivered underneath the scalp to numb the area of the incision. The skull was exposed, cleaned

with sterile cotton swabs and covered in a thin layer of cyanoacrylate adhesive (Krazy Glue, Elmer's Products, Atlanta, GA). After applying a coating of adhesive luting cement (C and B-Metabond, Parkell, Inc., Edgewood, NY) onto the layer of cyanoacrylate adhesive, the titanium head plate was lowered atop the skull and secured with additional application of luting cement. The headplate was centered about the body's anterior-posterior axis and equally spaced between bregma and lambda. For mice exposed to visual stimuli, however, head plate position was sufficiently posterior to prevent occlusion of the visual stimuli by the head plate. Mice were allowed at least one full week and typically greater than 4 weeks to recover before any testing was performed (*Table 1*). All animals were singly housed on a 12 hr/12 hr light/dark cycle and were tested during their dark phase.

## Design of the Virtual Burrow Assay (*Figure 1*)

The hardware design and control software are freely available for noncommercial use under the Creative Commons License at git.io/vA47E. The Virtual Burrow Assay (VBA) consists of a tube enclosure (virtual burrow) constructed out of a cardboard tube (looming experiments in *Figure 3*, and aversive learning experiments in *Figure 5*) or a 3D-printed polylactic acid tube (all other experiments, 45.5 mm inner diameter, 49 mm outer diameter, 7 cm long). In 3D-printed variants, the back of the tube is sealed and a trimmed absorbent underpad (Fisher Scientific, Hampton, NH) is affixed to the bottom. For the air puff (*Figure 2*), habituation (*Figure 4*) and aversive learning experiments (*Figure 5*) the tube included a 1 cm wide extension spanning approximately 1/3 of the tube's bottom circumference. In all designs, a 4 cm long, 0.5 mm diameter wooden rod is adhered to the front tip of the tube, 1 cm from the bottom, in order for animals to grip and rest their forelimbs. The diversity in tube material and geometry reflects the fact that assay design evolved concurrent with the sequence of experiments described in this manuscript. The final design of the tube (mark 21) can be consulted in the CAD folder on the GitHub repository: 3D-printed, including 1 cm extension (*Fink and Schoonover, 2018*; copy archived at https://github.com/elifesciences-publications/VBAcmd).

The virtual burrow is coupled to near-frictionless air bushings at 14 psi input pressure (New Way Air Bearings, Aston, PA), which slide along two rails in order to constrain movement to one dimension (rail design and assembly: T. Tabachnik, ZMBBI Advanced Instrumentation, Columbia University: fabrication: Ronal Tool Company, Inc., York, PA) (*Figure 1A*, right and *Figure 1B*). The animal is head-fixed via custom-machined stainless steel headplate holders (G. Johnson, Columbia University) that secure the titanium headplate. The entire VBA apparatus rests atop an adjustable platform (Thorlabs, Newton, NJ) to permit precise translation of the position of the tube with respect to the head. With t head thus secured, the animal's body rests freely inside the virtual burrow, its forepaws resting on the horizontal bar placed at the burrow's threshold, its hind limbs gripping the burrow's interior

A linear actuator (Part number: L12-30-50-12-I, Firgelli Automations, Ferndale, WA), tethered to the virtual burrow with fishing line (0.15 mm diameter nylon tippet, 4.75 pound test, Orvis, Sunderland, VT) constrains how far the animal may ingress into the burrow at any given time (*Figure 1B,C*). This parameter can be manually or programmatically varied over the course of the experiment. A force sensor (Futek FSH02664 load cell with Futek QSH00602 signal conditioner, Futek, Irvine, CA) reports whether, and how strongly, the animal is pulling against the linear actuator in its effort to ingress. Upon head-fixation in the VBA, mice invariably ingress as far as the linear actuator command position permits (*Figure 1C*). When the linear actuator retracts the burrow away from the ingress position (egress position, 10–20 mm posterior to ingress position), mice resist the translation, pulling against the tether in an effort to move the burrow back up around their body (*Figure 1D*). This effort typically generates between 0.4 and 1 N of force, corresponding in some cases to more than three times animal's own body weight (in grams-force). We have not observed any mice that fail to resist retraction of the virtual burrow.

A laser displacement sensor (Part number: ILD1302-50, Micro-Epsilon, Dorfbach, Germany) is aimed at a flag affixed to the horizontal bar that joins the air bearings in order to measure the linear displacement of the tube along its axis of motion. The readout of the laser displacement sensor yields a continuous, time-dependent, one-dimensional variable. It is this quantity – how far the animal has pulled the virtual burrow around its body – that tracks ingress in response to a given cue.

For all experiments reported here the analog voltage signals from the laser displacement sensor and the force meter were acquired and digitized at 10 kHz using a Cerebus Neural Signal Processor (Blackrock Microsystems, Salt Lake City, UT).

**Table 1.** Animals used in this study.

Identifying information for all animals used in this study.

| Animal number | Age at surgery | Age at expt. | Data | Notes |
|---|---|---|---|---|
| 2017022711 | 17 wks | 31 wks | Ingress, *Figure 1C,D* | - |
| 2017022712 | 17 wks | 31 wks | Ingress, *Figure 1C,D* | Also used for air puff expt. |
| 2017031302 | 10 wks | 22 wks | Ingress, *Figure 1C,D* | Also used for air puff expt. |
| 2017032101 | 11 wks | 22 wks | Ingress, *Figure 1C,D* | - |
| 2017032401 | 8 wks | 19 wks | Ingress, *Figure 1C,D* | - |
| 2017032402 | 11 wks | 22 wks | Ingress, *Figure 1C,D* | - |
| 2017033001 | 9 wks | 20 wks | Ingress, *Figure 1C,D* | Also used for air puff expt. |
| 2017033002 | 9 wks | 20 wks | Ingress, *Figure 1C,D* | Also used for air puff expt. |
| 2017042103 | 12 wks | 20 wks | Ingress, *Figure 1C,D* | - |
| 2017042105 | 12 wks | 20 wks | Ingress, *Figure 1C,D* | Also used for air puff expt. |
| 2018032602 | 8 wks | 14–16 wks | *Figure 1—figure supplement 1* | |
| 2018040204 | 9 wks | 15–17 wks | *Figure 1—figure supplement 1* | |
| 2018040206 | 9 wks | 15–17 wks | *Figure 1—figure supplement 1* | |
| 2018040208 | 9 wks | 15–17 wks | *Figure 1—figure supplement 1* | |
| 2018040212 | 9 wks | 15–17 wks | *Figure 1—figure supplement 1* | |
| 2016082207 | 11 wks | 18 wks | Air puff, *Figure 2* | *Figure 2B,C* from this animal |
| 2017022712 | 17 wks | 31 wks | Air puff, *Figure 2* | Also used for *Figure 1C,D* |
| 2017031302 | 10 wks | 22 wks | Air puff, *Figure 2* | Also used for *Figure 1C,D* |
| 2017033001 | 9 wks | 20 wks | Air puff, *Figure 2* | Also used for *Figure 1C,D* |
| 2017033002 | 9 wks | 20 wks | Air puff, *Figure 2* | Also used for *Figure 1C,D* |
| 2017042105 | 12 wks | 20 wks | Air puff, *Figure 2* | Also used for *Figure 1C,D* |
| 2018061501 | 7 wks | 15 wks | Air puff, *Figure 2* | - |
| 2018061201 | 7 wks | 15 wks | Air puff, *Figure 2* | - |
| 2018061202 | 7 wks | 15 wks | Air puff, *Figure 2* | - |
| 2018061301 | 7 wks | 15 wks | Air puff, *Figure 2* | - |
| 2016071401 | 15 wks | 21 wks | Visual stimuli, *Figure 3* | Expanding disk |
| 2016072401 | 16 wks | 21 wks | Visual stimuli, *Figure 3* | Expanding disk |
| 2016072702 | 17 wks | 21 wks | Visual stimuli, *Figure 3* | Expanding disk |
| 2016081901 | 10 wks | 11 wks | Visual stimuli, *Figure 3* | Receding disk |
| 2016081902 | 10 wks | 11 wks | Visual stimuli, *Figure 3* | Receding disk |
| 2016081903 | 10 wks | 11 wks | Visual stimuli, *Figure 3* | Receding disk |
| 2015111301 | 15 wks | 56 wks | Visual stimuli, *Figure 3* | Sweeping disk |
| 2015111602 | 12 wks | 52 wks | Visual stimuli, *Figure 3* | Sweeping disk |
| 2015111701 | 12 wks | 52 wks | Visual stimuli, *Figure 3* | Sweeping disk |
| 2016082302 | 11 wks | 18 wks | Odor habituation, *Figure 4A–D* | O1: Lim. O2: Oct. O3: Hex. |
| 2016082401 | 14 wks | 21 wks | Odor habituation, *Figure 4A–D* | O1: Oct. O2: Hex. O3: Lim. |
| 2016082402 | 14 wks | 21 wks | Odor habituation, *Figure 4A–D* | O1: Hex. O2: Oct. O3: Lim. |
| 2016082404 | 14 wks | 21 wks | Odor habituation, *Figure 4A–D* | O1: Hex. O2: Oct. O3: Lim |
| 2016082405 | 14 wks | 21 wks | Odor habituation, *Figure 4A–D* | O1: Oct. O2: Hex. O3: Lim. |
| 2017071603 | 13 wks | 34 wks | Odor habituation, *Figure 4E–H* | - |
| 2017071401 | 13 wks | 34 wks | Odor habituation, *Figure 4E–H* | - |
| 2017071605 | 13 wks | 34 wks | Odor habituation, *Figure 4E–H* | - |
| 2017071601 | 13 wks | 34 wks | Odor habituation, *Figure 4E–H* | - |

*Table 1 continued on next page*

Table 1 continued

| Animal number | Age at surgery | Age at expt. | Data | Notes |
|---|---|---|---|---|
| 2017071602 | 13 wks | 34 wks | Odor habituation, *Figure 4E–H* | - |
| 2016082006 | 11 wks | 21 wks | Aversive odor learning, *Figure 5, 6* | - |
| 2016082005 | 11 wks | 21 wks | Aversive odor learning, *Figure 5, 6* | - |
| 2016082003 | 11 wks | 21 wks | Aversive odor learning, *Figure 5, 6* | - |
| 2016082001 | 11 wks | 21 wks | Aversive odor learning, *Figure 5, 6* | - |
| 2016071401 | 15 wks | 30 wks | Aversive odor learning, *Figure 5, 6* | Also used for *Figure 3* |
| 2016081903 | 10 wks | 20 wks | Aversive odor learning, *Figure 5, 6* | Also used for *Figure 3* |
| 2016081902 | 10 wks | 20 wks | Aversive odor learning, *Figure 5, 6* | Also used for *Figure 3* |
| 2016081901 | 10 wks | 20 wks | Aversive odor learning, *Figure 5, 6* | Also used for *Figure 3* |
| 2016082602 | 14 wks | 23 wks | Aversive odor learning, *Figure 5, 6* | - |
| 2017092101 | 15 wks | 26 wks | Neurophysiology recording, *Figure 7* | |

DOI: https://doi.org/10.7554/eLife.45658.018

## Trial structure and closed loop control

Before each trial the control system pulls the virtual burrow back to the egress position and waits until the force measured by the force meter drops below a user-specified threshold, indicating that the animal has ceased to resist burrow retraction (*Figure 1—figure supplement 2*). The linear actuator is then advanced to the ingress position, slackening the tether and permitting free movement of the burrow. If the animal spontaneously ingresses prior to stimulus onset, as measured by the laser displacement sensor, the trial is aborted, the burrow is again retracted to the egress position, and the sequence repeats. Once the mouse has maintained the free, egress position without attempting to ingress within a specified duration, and has maintained the standard deviation of the tube position below a user-specified threshold for a specified delay period, the stimulus is delivered. During stimulus presentation, and a set duration following stimulus offset, the control system is switched to open loop, permitting the mouse to pull the burrow up to the ingress position if it wishes.

The burrow position (measured by the laser displacement sensor), burrow force (measured by the force sensor), and the servo position (state of the linear actuator) are analog inputs to a National Instruments card with analog and digital in/out (USB-6008, National Instruments, Austin, TX). The servo position is controlled by the same National Instruments card. (The position and force signals are simultaneously acquired on a separate DAQ, as described in the previous section.)

Prior to testing, naïve mice are head-fixed in the VBA and given 2–10 mn to acclimate to the contingencies in open loop (free movement of the burrow). Without exception, mice maintain the burrow in the ingress position throughout this acclimation period (*Figure 1C*). Then they are acclimated to the closed loop mode; after an initial period of sustained struggle to maintain the burrow in the ingress position (*Figure 1D*), mice cease resisting and eventually consent to holding the burrow in the egress position even after the linear actuator has advanced, slackening the tether and granting the mouse control over the burrow. The duration of the closed-loop acclimation period varied across mice in these experiments (1–20 mn) as the experimenters improved upon and acquired experience withthe assay; presently mice do not require more than ~5 mn total acclimation time (typically two mn for open loop and ~3 mn for closed loop). Trial blocks begin once the animal reliably holds the burrow in the egress position for >30 s between spontaneous ingresses. Trial initiation is delayed until after the mouse has held the burrow in the egress position with minimal movement for several seconds so as to ensure that the animal is in a comparable behavioral state prior to each trial.

## Air puff stimulus
### Characterization of ingress (*Figure 2A–C*)

Animals were head-fixed in the VBA and permitted to acclimate to head fixation for 2–4 mn with the VBA on open loop, after which the VBA was switched to the closed loop configuration; stimuli were administered once the animal readily gave trials (after approximately 1–2 mn). An 18-gauge, blunt

syringe needle delivered air puff stimuli to elicit either ingress (*Figure 2B*, needle tip 2 mm from snout, air pressure 80 psi, puff duration 200 msec, ITI 180 s, 15 trials per animal), or flinch (*Figure 2C*, needle tip 150 mm from nose, air pressure 2 psi, puff duration 20 msec). The example traces in *Figure 2B,C* depict responses to 15 consecutive strong air puff stimuli (2B) and one weak air puff stimulus (2C) delivered to a single, representative animal. The population statistics reported in the results section were measured in a separate cohort of 5 mice (three trials each) that had previously been exposed to odor stimuli in the course of an unrelated experiment in which they were administered a saline vehicle subcutaneously approximately 45mn prior to data acquisition.

## Tube/platform variants (*Figure 2D–F*)

We employed the same protocol as above, except using 250 msec puff duration and >90 s ITI, four animals per condition, three trials per animal; because mice occasionally bumped the tip of the blunt syringe needle when it was positioned close to their hindquarters, a distance of 10 mm was employed when puffing both the hindquarters and the snout. In order to deliver air puffs to the hindquarters a variant of the 3D-printed tube was designed such that the top portion (10 mm) of the tube's back 'wall' was left open. We note that it took great effort and time for animals to maintain a stable position on the open platform as they initially tended to thrash around and occasionally rotate their bodies at a > 45 degree angle with respect to the axis of their heads; this contrasts sharply with the apparently calm demeanor and stable posture characteristic of animals standing inside of tubes. Puffs were administered to the snout after animals had maintained a stable posture in which the body axis was aligned with the head's for >5 s.

To determine latency of the air puff, we measured the time between the TTL pulse controlling valve opening and the displacement of a small polystyrene weighing boat placed 2 mm distant from the blunt syringe needle (data not shown). We then subtracted the time between TTL pulse and measured displacement to determine the latency between TTL command and air puff stimulus at the nose. To account for variability in the position of the nose of the mouse with respect to the needle tip, we varied the precise location of the syringe needle over a range of distances similar to variability in distance between the syringe needle and the animal's nose across experiments. We observed negligible variability in latencies across this distance range.

For this and all experiments, a background of bandpass-filtered acoustic white noise (1000–45000 Hz; approximately 7 dB) was played throughout. The VBA apparatus was placed inside a custom-made sound attenuating chamber resting on an air table (TMC, Peabody, MA). All experiments were conducted under conditions of darkness, except when visual stimuli were presented, in which case the visual stimuli themselves provided the only source of visible illumination. The VBA was illuminated with infrared light to permit simultaneous video recording. For the experiments studying responses to visual stimuli, the chamber was open to accommodate the display screen but the lights in the room were off and the door was closed.

## Visual stimulus (*Figure 3*)

For experiments examining responses to visual stimuli, nine mice (three per condition) were acclimated to head fixation in the VBA for 3 mn in the open loop configuration. Following a subsequent 10-mn acclimation period with the VBA in the closed loop configuration, the animal was again permitted to freely ingress in open loop for 3–5 mn. The VBA was then returned to the closed loop configuration and once the animal did not spontaneously ingress for periods greater than 30 s (typically after approximately 1–3 mn) visual stimuli were delivered.

The visual stimuli employed were based on those described in *De Franceschi et al. (2016)*. Briefly, the stimuli were presented on a Dell 1707FP 17' LCD monitor, 1280 × 1024, 60 Hz, elevated 30 cm and centered above the animal's head. The three stimuli, generated using the Psychophysics Toolbox Version 3 in MATLAB (Mathworks, Natick, MA), consisted of a black disk presented against a gray background: expanding disk ('loom'), widening from 2° to 50° over 250 msec, holding the 50° disk for 500 msec; contracting disk ('recede'), diminishing from 50° to 2° over 250 msec, holding the 2° disk for 500 msec; and sweeping disk ('sweep'), a 5° disk sweeping smoothly across the diagonal of the screen at a rate of 21°/sec. In order to permit synchronization of stimulus timing with burrow position measurement, the software controlling the visual stimulus also controlled a PWM signal (generated by an Arduino Uno, Adafruit, New York, NY; acquired as an analog voltage input

digitized at 10 KHz simultaneous to the position and force signals) that encoded the identity and timing of the visual stimuli.

We divided nine mice into three groups of three animals, one group per stimulus type, and presented each mouse only one of the stimulus types in a single session of five stimulus presentations separated by a 10-mn ITI. The data for each stimulus type are pooled across animals for each group.

## Odorant stimuli

In all experiments involving odorant stimuli we avoided using molecules known to elicit systematic attraction or aversion in freely moving animals, such as trimethyl-thiazoline, a volatile cue secreted from the anal gland of fox that elicits avoidance and fear responses (*Hebb et al., 2004*) or 2-phenyl-ethanol, a component of rose oil that elicits attraction (*Root et al., 2014*).

We used a custom-built olfactometer to deliver odorant stimuli. Briefly, a nose port constructed of polyether ether ketone (PEEK) was placed approximately 1 mm away from the animal's nose. When no odorant stimulus was given, the port delivered a steady stream of air (one liter per minute, controlled by a mass flow controller, GFCS-010201 from Aalborg, Orangeburg, New York) that had bubbled through a 50 ml glass bottle containing 15 ml dipropylene glycol (DPG, Part number: D215554, Sigma-Aldrich, St. Louis, MO). To deliver an odorant stimulus, a four-way valve (Part number: LSH360T041, NResearch Inc., West Caldwell, NJ) routed the air stream to exhaust, replacing it with a stream of odorized air; the odorant stimulus was switched off by the four-way valve routing the odorized air back to exhaust. Monomolecular odorants (cis-3-Hexen-1-ol, catalog number W256307; (R)-(+)-Limonene, catalog number 183164; Octanal, catalog number O5608; Ethyl trans-3-hexenoate, catalog number W334200, all from Sigma-Aldrich, St. Louis, MO) were dissolved in 15 ml DPG at a concentration of 2% volume/volume for Hexenol, Limonene, and Octanal and 4% volume/volume for Ethyl trans-3-hexenoate in separate 50 ml glass bottles. After passing through the nose port all gas was routed to a photo-ionization detector (miniPID, Aurora Scientific, Aurora, ON, Canada) to permit constant monitoring of odorant concentration. To avoid contamination, all material in contact with the odorized air stream was constructed in either Teflon, Tefzel, or PEEK. The flow of the air and odor streams were equalized before each experiment (using mass flow meter GFMS-010786 from Aalborg, Orangeburg, NY) and the tubing carrying the two streams from the four way valve was set to equal length and impedance to minimize variation in flow rate upon switching between the air and odor streams.

## Odor habituation (*Figure 4*)

For odor habituation experiments, five mice were head-fixed in the VBA and allowed to acclimate in the open loop configuration for 5 mn. In order to promote all-or-nothing behavioral responses, we reduced the input pressure to the air bushings from the customary 14 psi to 2 psi, thereby requiring the animal to generate greater force to initiate an ingress than under near-frictionless conditions. After acclimation the VBA was set to closed loop for 10–15 mn. Following acclimation to the closed-loop mode, the animal was then presented with odorant stimuli with the VBA in the closed loop mode. A single odorant, Odor 1 was presented 15 times. Then, a second odorant, Odor 2, was introduced and the two odorants were presented 15 times each, pseudo-randomly interleaved within blocks in which each of the two odorants was presented in every block. Finally, a third odorant, Odor 3, was added and all three odorants were presented in 15 final blocks of three trials each. Each odorant stimulus was presented once per block. All odorant stimuli were 8 s in duration and the ITI was 40 s. Limonene, Octanal, and Hexenol were used as odorant stimuli with different animals receiving different odorants for the Odor 1, Odor 2, and Odor three stimuli (*Table 1*). Because we refrained from using odors known to elicit innate attraction or aversion such as trimethyl-thiazoline, or 2-phenylethanol, it is unlikely that the egress and ingress responses we observed in early trials were due to the odorants having an innately attractive or aversive quality beyond their novelty.

We noted that we did not observe renewed ingress to familiar odorants following presentation of unexpected ones. This absence of dishabituation (*Groves and Thompson, 1970*; *Thompson and Spencer, 1966*) may reflect the use of lower input pressure to the air bushings in this experiment, which promotes all-or-nothing responses by increasing friction between the bearings and the rails but also dramatically decreases the sensitivity of the assay.

For the exploration experiments, the VBA control software was set to open loop mode and the fixed odor port was replaced with an odor port coupled to the virtual burrow, such that the animal was required to egress approximately 30 mm from the fully ingressed position to touch the port with its nose. The odorant stimuli (Ethyl trans-3-hexenoate) were 8 s in duration and the ITI was 60–120 s. In the rare instances during which the animal was not ingressed at the conclusion of the ITI, stimulus delivery was delayed until the animal had resumed a fully ingressed position.

## Odor-shock conditioning and testing (*Figures 5,6*)

### 1: Pre-test

The nine mice used in odor-conditioning experiments were first habituated to the three odorant stimuli employed (CS+: Limonene, CS-: Hexenol, Odor3: Octanal). Animals were placed in the VBA and acclimated to head fixation for 5 mn with the VBA in open loop, after which the VBA was switched to closed loop for 10 mn. Following the 10-mn closed loop acclimation period, the VBA was restored to the open loop configuration for 5 mn to permit the animal to freely ingress before testing, and then returned to closed loop immediately before commencing odorant stimulus delivery. Odorant stimuli (8 s duration) were presented in 10 blocks of three pseudo-randomly interleaved trials (60 s ITI) such that each stimulus was presented once per block. Following completion of the 10 stimulus blocks, animals were immediately removed from the VBA and returned to their home cage. Air pressure in the air bearings was set to 3 psi during habituation.

### Day 2: Conditioning

Conditioning was performed one day after odor habituation. A fear conditioning chamber (14.2 mm wide, 16.2 mm long, 12.6 mm high, Med Associates, Fairfax, VT) was employed. Under conditions of darkness with an acoustic background of white noise, mice were placed inside the fear conditioning chamber on the open, gloved hand of the experimenter. Once the animal had freely entered the fear conditioning chamber, the door was closed and the animal was allowed to acclimate for 5 mn. Eight blocks of CS+ and CS- odorant stimuli were presented in pairs of pseudo-randomly interleaved trials. The odorant stimuli were 10 s in duration with a 5-mn ITI. During the final 2 s of presentation of the CS+ stimulus only, the floor of the fear conditioning chamber was electrified (intensity 0.70–0.73 mAmp). Upon completion of all eight trials, the mouse was permitted to recover for 5 mn in the fear conditioning chamber and then returned to its home cage.

### Day 3: Test

One day after conditioning animals were returned to the VBA to test responses to all odorant stimuli. Test was identical to pre-test except that eleven stimulus blocks were presented and air pressure in the VBA air bearings was set to 15 psi.

## Neurophysiological recording (*Figure 7*)

Electrophysiology was performed using silicon probes (A1 × 32-Poly3-5mm-25s-177-H32_21 mm, NeuroNexus, Ann Arbor, MI) chronically implanted into anterior piriform cortex (APC). At least one week following headplate attachment, mice were anesthetized with ketamine/xylezine and a craniotomy centered 1,150 μm posterior to the rostral confluence of the dorsal sinuses and 2,250 μm lateral to the midline was performed using a dental drill (Osada Success 40, Osada Electric Company, Tokyo, Japan). The probe was lowered until the dense spiking characteristic of layer 2 of APC was detected, at which point it was cemented in place. Animals were allowed to recover for at least four weeks before any recording took place to allow for the tissue to fully settle around the implanted probe.

Neural signals were acquired simultaneous to behavior using a Cerebus Neural Signal Processor (Blackrock Microsystems, Salt Lake City, UT). Custom written scripts in Matlab were then used to filter and preprocess the data, which were then automatically spike sorted using Kilosort (*Pachitariu et al., 2016*) and manually curated using phy (https://github.com/kwikteam/phy).

## Statistics

To determine whether responses in the VBA differed across experimental conditions, we asked whether the likelihood of ingress was larger in one condition than another. For the purposes of this

test we define an ingress as a maximum change in burrow position greater 2 mm (the results of the statistical tests are robust to the choice of threshold; see *Figure 5—figure supplement 2*). For each condition we pooled all ingress responses across mice and used a two-proportion z-test with the null hypothesis that the probability of ingress in the tested condition was less than or equal to the probability of ingress in the other condition. In figures, one star (*) indicates $p<0.05$, two stars (**) indicate $p<0.01$, and three stars (***) indicate $p<0.001$.

## Acknowledgments

We are grateful to Tanya Tabachnik for the design and fabrication of the frictionless rail; Hopi E Hoekstra and Caroline K Hu for helpful conversations regarding burrowing behavior, as well as for providing unpublished video of mice in the wild; Ya-tang Li, Markus Meister and Samuel G Solomon for advice on flight-inducing visual stimuli; John P Cunningham for advice on statistical analyses; Gary W Johnson for precision machining; J Andrew Miri for suggesting the use of a frictionless rail; Paul R Stegall and Damiano Zanatto for advice on mechanical engineering; Alberto Hernandez for supplying cardboard tubes; Annegret L Falkner for Friday night science; D Caroline Blanchard, Rui M Costa, Walter M Fischler, John W Krakauer, Eve Marder, and Catherine H Rankin for critical comments on the manuscript; Ethan S Bromberg-Martin, Michael E Goldberg, and Alla Y Karpova for insightful comments; and the 2016 Janelia Junior Scientist Workshop on Neural Circuits and Behavior.

## Additional information

### Funding

| Funder | Author |
| --- | --- |
| Howard Hughes Medical Institute | Richard Axel |
| Helen Hay Whitney Foundation | Andrew JP Fink |

The funders had no role in study design, data collection and interpretation, or the decision to submit the work for publication.

### Author contributions

Andrew JP Fink, Carl E Schoonover, Conceptualization, Software, Formal analysis, Investigation, Methodology, Writing—original draft, Writing—review and editing; This paper is the result of a close collaboration between AJPF and CES, who made equal contributions to the project; Richard Axel, Resources, Formal analysis, Supervision, Funding acquisition, Writing—review and editing

### Author ORCIDs

Andrew JP Fink [iD] http://orcid.org/0000-0003-4191-3298
Richard Axel [iD] http://orcid.org/0000-0002-3141-4076
Carl E Schoonover [iD] https://orcid.org/0000-0002-6397-1010

### Ethics

Animal experimentation: All of the animals were handled according to approved institutional animal care and use committee (IACUC) protocols AC-AAAI8650 and AC-AAAT5466 of Columbia University Medical Center. All surgery was performed under Isoflurane anesthesia, and every effort was made to minimize suffering.

### Decision letter and Author response

Decision letter https://doi.org/10.7554/eLife.45658.021
Author response https://doi.org/10.7554/eLife.45658.022

## Additional files

### Supplementary files
• Transparent reporting form
DOI: https://doi.org/10.7554/eLife.45658.019

### Data availability
The dataset is available as part of a public GitHub repository (https://github.com/goatsofnaxos/VBAcmd) that details specifications for the assembly of the device and source code for the software that controls it (copy archived at https://github.com/elifesciences-publications/VBAcmd).

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
