## [Decision Letter]

[Editors’ note: a previous version of this study was rejected after peer review, but the authors submitted for reconsideration. The first decision letter after peer review is shown below.]

Thank you for submitting your work entitled "A naturalistic assay of habituation, discrimination, exploration and avoidance at millisecond precision in head-fixed mice" for consideration by *eLife*. Your article has been reviewed by four peer reviewers, including Peggy Mason as the Reviewing Editor and Reviewer #1, and the evaluation has been overseen by a Senior Editor. The following individuals involved in review of your submission have agreed to reveal their identity: Venkatesh N Murthy (Reviewer #2); Michael Brecht (Reviewer #4).

Our decision has been reached after consultation between the reviewers. Based on these discussions and the individual reviews below, we regret to inform you that your work will not be considered further for publication in *eLife*.

The reviewers agreed that the setup could be useful but they also remained unconvinced that the behavior exhibited by the mice represents burrow ingress and egress. One simple alternative explanation is that the mouse is backing away from a noxious or aversive stimulus and approaching to explore a novel stimulus.

Reviewer #1:

This is a clever method to look at ethologically relevant behavior in a setup highly conducive to neurophysiological recordings. The setup is simple, the mice perform easily and without training and the timing is amenable to synaptic circuit analysis.

Two suggestions for short (1-3 sentence) discussions that could add value to the reader. 1) Is there a reason why an extension of the top of the tube was not used that would cover the head? Put another way, the natural behavior is diagonally oriented and involves head covering in the axis of the body whereas this setup is horizontal and does not cover the head. 2) The latency to ingress to puff 18-19 ms compared to that of CS+ presentation ~700 ms is fascinating. A comment or two on the supporting circuits would be helpful. Is this latency compatible with the amygdala-involving circuits that have been sketched out? It would seem a bit long. In any case, this latency difference constrains the conditioning circuits to something beyond oligosynaptic.

Reviewer #2:

In this paper Fink et al. describe a novel apparatus which allows mice to exhibit burrowing-like behavior while head-fixed. The authors demonstrate that mice quickly learn to push or pull in order to cause a tube surrounding their body to come forward to more fully cover their body or to go backward to expose their body. They demonstrate that the mice do this movement very quickly and in a stereotyped manner. They also show that mice tend to treat the tube as if it was a naturalistic burrow (although the mice themselves are not moving), by "burrowing" in response to airpuff and in response to a looming visual stimulus of increasing size. They then demonstrate that when the position in the tube is coupled with odor intensity to imitate the action of exiting the burrow to explore an odor source or retreating back away from the odor source, that the mice tend to exhibit exploratory behavior by occasionally "exiting" to explore new odors, but quickly becoming habituated to them. Finally, the authors also condition the mice to associate one odor with a foot shock and demonstrate that the mice tend to "retreat" from this odor as compared to an odor which was not coupled with foot shock. Overall, this behavioral assay seems to work well and is a nice addition to the behavioral assays that can be performed on head-fixed mice. However, I'm not convinced that this method in itself is an advance significant enough to warrant publication in *eLife*.

It's unclear how much of the behavior that the authors demonstrate is a result of the tube/burrow. For example, head-fixed mice tend to attempt to retreat from aversive stimuli like an airpuff even in the absence of the tube (easily seen and quantified from videos, for example). If the authors could compare with tube and without tube conditions and demonstrate that the tube makes a difference this would be more convincing. It seems important to add controls experiments to prove that the mouse's behavior is indeed a burrowing behavior, and not just a retreat response. This could be implemented, for example, by removing the top half of the tube and keeping everything else the same.

In general, while I agree that the time resolution of measuring the behavior readout could be an advantage over other behaviors such as freezing. But I fail to see its advantage over the measurement of other fast readouts, such as measuring blinking and limb motions with a high-speed camera, or measuring retreating movement on a moving belt.

Reviewer #3:

Fink et al. develop a head fixed apparatus for measuring mouse response to a variety of stimuli such as air puff, looming stimulus and odor presentations. Depending on whether the animals move forward or backward, they will either push or pull a tube (the "burrow") away from or around their body. The authors demonstrate that many of the behaviors probed do not require extensive training or acclimation. The apparatus allows for millisecond precision measurements of innate animal responses to stimuli in a way that is compatible with "standard electrophysiological and optical methods". While there are some novel aspects to the study, some major points limit my enthusiasm.

1) It is unclear if the "burrow" is at all required for the behaviors and animal responses probed here. Without a comparison to a situation without a burrow, how can the authors argue that this is a virtual burrowing assay? For example, if the authors simply placed a mouse on a platform (without a cover) or on a treadmill, could they have measured the same responses to stimuli? To put it another way, how do the authors argue against the idea that mice are not displaying ingress to a burrow, but are instead just backing up away from an aversive stimulus?

2) Sharp transitions in behavioral state will likely lead to significant brain motion that could preclude measurements with "standard electrophysiological and optical methods". The authors do not provide references for the methods to which they are referring. Assuming they are referring to high resolution imaging, single unit recording, and intracellular recording (among other methods), then these movements could substantially limit recording compatibility. This needs to be explicitly addressed.

3) A large suite of head-fixed behaviors already exist and it is not clear in the current manuscript how the "burrowing" assay compares and contrasts to these. For example, head-fixed odor assays (see for example research from the Rinberg or Bozza labs), numerous head-fixed airpuff response assays, and even head-fixed olfactory virtual reality behaviors measuring odor valence already exist (Radvansky et al).

4) The following requires quantification: "When placed inside the virtual burrow, mice invariably pull the tube up around themselves as far as possible-a behavior we define as "ingress" (Figure 1A, right). When granted full control over the virtual burrow, mice prefer to remain ingressed…"

Reviewer #4:

In their paper 'A naturalistic assay of habituation, discrimination, explorationand avoidance at millisecond precision in head-fixed mice' Fink et al. describe a virtual burrow set up for neuroscientific experiments. The authors convincingly argue that the virtual burrow system is a simple arrangement that exploits natural response tendencies of mice. The authors describe the system and then demonstrate that air puffs and looming visual stimuli trigger 'ingress' responses as such stimuli would in natural burrows. They also demonstrate olfactory behaviors and conditioning effects. The apparatus the authors describe is rather simple and I initially wondered if the findings constitute enough novelty for an *eLife* paper. Because so many investigators work with head-fixed mice, the potential usefulness of the technique and observations wins out over these other concerns, however. Thus, I am in principle supportive of this study. The presentation needs to be improved along the lines specified below.

Improving the presentation:

Figure 1

Figure 1B is most important panel of all figures and the paper. It looks good, but is not very helpful and far from ideal. It does not become clear here, which parts slide and how the mouse can egress and ingress. Equally disturbing is that other parts that are specified in the rather complex schematic in Figure 1C, i.e. the odor delivery, cannot be found here. I suggest enlarging panel 1B and adding explanations etc. Figure 1C, D should be decomplexified, we are not talking about Hodgkin and Huxley here, but about a rather simple behavioral control system.

Figure 2

Figure 2B, what exactly is plotted on the Y-axis? I did not get panel 2C from the figure or reading the legend, what exactly is plotted on the Y-axis? Please clarify.

Figure 3

Figure 3 is nice. It might be helpful to repeat the terms loom, recede, sweep also in panel A.

Figure 4

Figure 4E: The visual impression I get from Figure 4E is that the animals is egressing much more than it is ingressing. If I got the rest of the figure correctly the opposite is true, so the schematic is not ideal.

Figure 4F: Along the same line, maybe it would be good to indicate a 'zero/starting position' as a dashed line.

Figure 5

Figure 5 is rather busy, yet the findings are straightforward. Simplify.

[Editors’ note: what now follows is the decision letter after the authors submitted for further consideration.]

Thank you for submitting your article "Virtual burrow assay for headfixed mice measures habituation, discrimination, exploration and avoidance without training" for consideration by *eLife*. Your article has been reviewed by three peer reviewers, and the evaluation has been overseen by a Peggy Mason as Reviewing Editor and Catherine Dulac as the Senior Editor. The following individuals involved in review of your submission have agreed to reveal their identity: Venkatesh N Murthy (Reviewer #2); Michael Brecht (Reviewer #3).

The reviewers have discussed the reviews with one another and the Reviewing Editor has drafted this decision to help you prepare a revised submission.

Thank you for the responsive revisions. I would ask that you consider reviewer 1's concerns and add text to address these issues, particularly the issues of repeated use of the paradigm over multiple days and a comparison to previous head fixed olfaction methods. Reviewer 1 also suggests that adding in a platform condition with a back puff would make the control more complete. Please address this with either more experiments or a brief reason why more experiments would not enlighten.

Reviewer #1:

In their revision, Fink et al. present a substantially improved manuscript through the addition of data demonstrating that the burrow itself is actually needed for the VBA (Figure 2D-F). The addition of silicon probe single unit recording data during sharp movements is also a major improvement (Figure 7). These improvements address my previous main concerns. After reading the new manuscript, I have a few more concerns that I feel should be addressed before publication:

1) The authors demonstrate that no training is required for the VBA, however, in order for the apparatus to be of maximal use in the community, the behaviors probed with the VBA should persist for many sessions/days. Figure 5 begins to get at this question (same mice over 2 different days), but I feel that this point should be addressed more directly; this could be accomplished with some added discussion, or with data that the authors may already have showing other VBA behavioral responses over days with the same mice.

2) Figure 2E, F: for completeness, the puff back condition should be included for the platform case.

3) Introduction, second paragraph: Since one major application of the VBA appears to be to olfactory responses (Figures 4 and 5), it seems appropriate to cite and discuss some previous methods using olfactory stimuli in head-fixed mice here: Lovett-Barron, 2014; Radvansky, 2018; Resulaj, 2015.

Reviewer #2:

I don't have any technical comment for this revision because I think the behavior and setup are now well characterized.

The control experiments the authors performed definitely help to support the claim that the mice are performing a burrowing-like behavior rather than just backing away from the aversive stimulus. With the other advantages of this behavior (no need for pre-training, reliable responses, fast latency) which the authors describe, we can conclude that this behavioral setup is likely a useful tool addition to many other behavioral measures already in use (eye blink conditioning, freezing, licking, sniffing, virtual reality navigation, and so on). I continue to wonder whether developing a behavioral paradigm, with little by way of new neural or behavioral insight, is a significant advance in itself for publication in *eLife* – but I will be glad to leave the decision to the editors.

Reviewer #3:

The paper improved substantially. I think this will be a useful contribution given the widespread use of the head-fixed mouse preparation. The authors addressed my concerns. I support publication.

---

## [Author Response]

[Editors’ note: the author responses to the first round of peer review follow.]

The principal reason given for rejecting the manuscript was that the reviewers “remained unconvinced that the behavior exhibited by the mice represents burrow ingress and egress.” You cautioned that instead the mouse may simply be “backing away from a noxious or aversive stimulus and approaching to explore a novel stimulus”. This concern spurred us to perform further experiments that directly test this alternative explanation.

We asked:

1) Does ingress correspond simply to movement away from an aversive stimulus or does it constitute retreat within the burrow enclosure?

2) Do mice behave on a moveable, open platform as they do within the virtual burrow?

As we now describe in the subsection “Air puffs elicit rapid and reliable ingress”, and in a revised and expanded Figure 2, we conducted two experiments to address these questions.

Reviewer #2:[…] Overall, this behavioral assay seems to work well and is a nice addition to the behavioral assays that can be performed on head-fixed mice. However, I'm not convinced that this method in itself is an advance significant enough to warrant publication in eLife.It's unclear how much of the behavior that the authors demonstrate is a result of the tube/burrow. For example, head-fixed mice tend to attempt to retreat from aversive stimuli like an airpuff even in the absence of the tube (easily seen and quantified from videos, for example). If the authors could compare with tube and without tube conditions and demonstrate that the tube makes a difference this would be more convincing. It seems important to add controls experiments to prove that the mouse's behavior is indeed a burrowing behavior, and not just a retreat response. This could be implemented, for example, by removing the top half of the tube and keeping everything else the same.

We have performed two control experiments to address this concern (Figure 2D-F). What we found is that the presence of the burrow-like enclosure is critical for animals to exhibit the ingress behavior we have reported.

We first asked what happens if air puff is delivered behind the animal rather than in front, to its snout. The rationale was that if the animal were simply moving away from an aversive stimulus, then they should egress in response. Instead, we found that animals ingress even though the air puff is coming from behind them; had they not been headfixed, this would have resulted in movement *towards* the noxious stimulus. In other words, the drive to ingress is sufficiently strong as to overrides the drive to move away from the source of the aversive stimulus.

We then asked whether the burrow itself is necessary to observe ingress in response to air puff. (This experiment was suggested by you and reviewer #3.) For this we placed the animals on a moveable flat platform, rather than within a tube enclosure. The animals’ responses to air puff were erratic under these circumstances: following a transient “flinch” they did not move the platform in a consistent direction, or nearly as far as they did the enclosure. Therefore, the burrow itself is critical for observing ingress responses.

Together these experiments indicate that the ingress we observe in the VBA is consistent with retreat inside the burrow like enclosure, rather than the alternative explanation that this is “just a retreat response” from the source of the air puff.

Reviewer #3:[…] 1) It is unclear if the "burrow" is at all required for the behaviors and animal responses probed here. Without a comparison to a situation without a burrow, how can the authors argue that this is a virtual burrowing assay? For example, if the authors simply placed a mouse on a platform (without a cover) or on a treadmill, could they have measured the same responses to stimuli? To put it another way, how do the authors argue against the idea that mice are not displaying ingress to a burrow, but are instead just backing up away from an aversive stimulus?

We have performed two control experiments to address this concern (Figure 2D-F). What we found is that the presence of the burrow-like enclosure is critical for animals to exhibit the ingress behavior we have reported.

We first asked what happens if air puff is delivered behind the animal rather than in front, to its snout. The rationale was that if the animal were simply moving away from an aversive stimulus, then they should egress in response. Instead, we found that animals ingress even though the air puff is coming from behind them; had they not been headfixed, this would have resulted in movement *towards* the noxious stimulus. In other words, the drive to ingress is sufficiently strong as to overrides the drive to move away from the source of the aversive stimulus.

We then asked whether the burrow itself is necessary to observe ingress in response to air puff. (This experiment was suggested by you and reviewer #2.) For this we placed the animals on a moveable flat platform, rather than within a tube enclosure. The animals’ responses to air puff were erratic under these circumstances: following a transient “flinch” they did not move the platform in a consistent direction, or nearly as far as they did the enclosure. Therefore, the burrow itself is critical for observing ingress responses.

Together these experiments indicate that the ingress we observe in the VBA is consistent with retreat inside the burrow like enclosure, rather than the alternative explanation that they are “just backing up away from an aversive stimulus”.

2) Sharp transitions in behavioral state will likely lead to significant brain motion that could preclude measurements with "standard electrophysiological and optical methods". The authors do not provide references for the methods to which they are referring. Assuming they are referring to high resolution imaging, single unit recording, and intracellular recording (among other methods), then these movements could substantially limit recording compatibility. This needs to be explicitly addressed.

We have added a figure (Figure 7) showing stability of neurophysiological recording of single units using silicon probes taken simultaneous to sharp transitions in behavioral state in the VBA. We have also included in the main text a personal communication from a colleague reporting similar success with measurement of Calcium transients using 2photon microscopy through a GRIN lens.

3) A large suite of head-fixed behaviors already exist and it is not clear in the current manuscript how the "burrowing" assay compares and contrasts to these. For example, head-fixed odor assays (see for example research from the Rinberg or Bozza labs), numerous head-fixed airpuff response assays, and even head-fixed olfactory virtual reality behaviors measuring odor valence already exist (Radvansky et al).

The vast majority of currently employed behavioral assays for head-fixed mice require prior training in the sensorimotor contingencies of the instrument (for a simple case, see for e.g. Guo et al., 2014); even learning how to locomote on a virtual-reality treadmill while head-fixed (Dombeck et al., 2010) requires several days of acclimation and training; in practice it is not uncommon to discard animals that are not able to perform adequately given the contingencies (for e.g. in the Radvansky et al. study that cited, the authors attempted to train 18 mice but only 9 showed "acceptable behavior"). Given that current head-fixed assays depend upon learned behaviors to report sensory or cognitive quantities, and given that task proficiency is highly variable across animals, it is reasonable to expect that this variability would contaminate the measurement of even simple quantities such as sensory detection thresholds.

Furthermore, training engages circuitry that may be unrelated to the sensory or cognitive faculty under study, such as: modules that signal satiation, reward, or punishment; modules that implement the learning of the task's structure; and modules that implement the learning, planning, and execution of motor skills. Thus training introduces several layers of complexity in between the sensory or cognitive operations that are the object of study and the behavior that is observed in order to infer their properties. These additional layers are likely to further muddy the already challenging interpretation of lesion or perturbation studies (Wolff and Ölveczky, 2018), or of the relationship between neural activity and concomitant behavior.

In contrast, we have demonstrated that the VBA can measure a diversity of behaviors, ranging from innate avoidance and fear conditioning to stimulus discrimination and exploration, without the need for any training, or even prior acclimation to head-fixation. Indeed most of the data reported in this manuscript was acquired within minutes of the animals being head-fixed for the first time in their lives (while awake), and the fact that no animals were excluded from analysis speaks to the behavioral stereotypy we encountered. Naturally, we are not claiming that this device affords direct observation of the function of sensory and cognitive modules--but by tapping into behaviors that all animals produce "out of the box" instead of relying on training the VBA stands to dramatically reduce the complexity of the system necessary to produce meaningful behavior under head fixation.

In addition there are several other key ways in which the VBA represents an advance over existing behavioral assays for head-fixed mice.

- As an assay of Pavlovian fear conditioning, the VBA provides a precise time stamp of the onset of the conditioned response. To date, the most successful measurement of fear conditioning in head-fixed mice has resulted from the translation of conditioned lick-suppression paradigms (Bouton and Bolles, 1980, Lovett-Barron et al., 2014). However, in both freely moving and head-fixed mice, this method lacks fine temporal resolution, and moreover the behavioral readout is significantly more complex—the result of an interaction between an operant behavior (licking, which has to be trained) and a conditioned response.

- The VBA’s sensitive readout of motor output permits isolation of distinct, subtle behavioral motifs. This is exemplified, for example, in the resolution of “tremble” responses to conditioned odor stimuli (Figure 5—figure supplement 1) or the precise measurement of core and limb strength exerted by the mice (such as the force they exert in their attempts to resist the action of the linear actuator)—neither of which have been reported in head-fixed mice, to our knowledge.

- The selective ingress responses to looming visual stimuli we report in Figure 3 illustrate how ingress in the VBA can be used to study innate flight-like behavior. We are not aware of any head-fixed assay for responses to looming visual stimuli in head-fixed mice—or, for that matter, flight-like responses in general. You highlighted a recently published assay (Radvansky et al., 2018) that measures avoidance of innately aversive odorant stimuli; this approach contrasts with ours in that (1) it appears to measure avoidance rather than flight and (2) requires training in the sensorimotor contingencies of the assay.

- The VBA provides a sensitive assay of habituation to olfactory stimuli (Figure 4). While there exist head-fixed assays for habituation to novel odors (e.g. Verhagen et al., 2007), these rely on measurement of the sniff rate. However in our hands this paradigm has proven unreliable and noisy; in fact, part of the motivation for developing the VBA was our dissatisfaction with this method. Using ingress (4BC) rather than sniff rate has yielded results that are substantially cleaner and consistent across animals. Selective ingress to novel odors (Figure 4B-C) can be used as a sensory detection/discrimination assay without requiring the extensive training typically required for these kinds of measurements (cf. Guo et al., 2014). We note that here training introduces an additional complexity: as the animals gain proficiency in the task, this is likely to result in perceptual learning (Fahle and Poggio, 2002), causing overestimation of default perceptual performance.

- The configuration of the assay described in Figure 4E-H, in which the odor source is coupled to the tube itself, requires the animal to override its strong drive to stay inside the enclosure in order to better sample the stimulus outside. To the best of our knowledge, there are no reports of the measurement of exploration in headfixed mice.

A summary of this response to your concern has now been included in the text. In particular, the Introduction has been rewritten so that the focus is on the principal difference between the VBA and the large suite of already existing head-fixed behavioral assays (training) and the discussion compares and contrasts the VBA with other existing assays.

4) The following requires quantification: "When placed inside the virtual burrow, mice invariably pull the tube up around themselves as far as possible-a behavior we define as "ingress" (Figure 1A, right). When granted full control over the virtual burrow, mice prefer to remain ingressed…"

We have added a panel (Figure 1C) that plots example traces and quantifies position during open-loop mode (habituation). This comment prompted us to also add a second panel (Figure 1D) that quantifies measurements of the resistance animals exhibit when they are pulled out of the tube.

Reviewer #4:[…] The presentation needs to be improved along the lines specified below.Improving the presentation:Figure 1Figure 1B is most important panel of all figures and the paper. It looks good, but is not very helpful and far from ideal. It does not become clear here, which parts slide and how the mouse can egress and ingress. Equally disturbing is that other parts that are specified in the rather complex schematic in Figure 1C, i.e. the odor delivery, cannot be found here. I suggest enlarging panel 1B and adding explanations etc. Figure 1C, D should be decomplexified, we are not talking about Hodgkin and Huxley here, but about a rather simple behavioral control system.

We have reworked Figure 1 and the associated text to present the assay more clearly; we have cut 1C and 1D, moving them to a figure supplement, in order to focus on the core features of the assay; and we have added two videos (Figure 1—videos 1 and 2) to illustrate its operation.

Figure 2Figure 2B, what exactly is plotted on the Y-axis? I did not get panel 2C from the figure or reading the legend, what exactly is plotted on the Y-axis? Please clarify.

Y-axes are now labeled as “tube position (mm)”. Legend for 2C reads: “Example of flinch in response to weak air puff. Downward going, approximately 2-Hz oscillations correspond to the animal’s breathing cycle. Upward going low-amplitude, transient deflection corresponds to startle in response to air puff (grey box, 20 msec, 2 psi).” Text for 2C reads: “It was also possible on occasion to induce a flinch, rather than ingress, by delivering a very weak air puff (2 psi, 15 cm distance). This apparent startle response (Davis, 1984) is characterized by a transient change in burrow position clearly distinct from the ongoing movement of the burrow caused by the animal’s breathing.”

Figure 3Figure 3 is nice. It might be helpful to repeat the terms loom, recede, sweep also in panel A.

This is now done.

Figure 4Figure 4E: The visual impression I get from Figure 4E is that the animals is egressing much more than it is ingressing. If I got the rest of the figure correctly the opposite is true, so the schematic is not ideal.

This visual impression is correct: ingress magnitudes range from 0mm to approx. 15mm (Figure 4D) and egress magnitudes range from 0mm to approx. -30mm (Figure 4H). We have modified the schematic to prevent confusion (see next comment).

Figure 4F: Along the same line, maybe it would be good to indicate a 'zero/starting position' as a dashed line.

This is now done. In addition we have drawn corresponding dashed lines in the diagrams in 4E, and that panel’s legend now includes: “Note that, as above, 0mm corresponds to the virtual burrow’s position prior to stimulus presentation; however in these experiments the animals began each trial in the ingress position rather than in the egress position.”

Figure 5Figure 5 is rather busy, yet the findings are straightforward. Simplify.

In order to simplify Figure 5 and simultaneously give the latency result more prominence we have removed panels J-L, moving them to a separate new figure (Figure 6).

[Editors' note: the author responses to the re-review follow.]

The reviewers have discussed the reviews with one another and the Reviewing Editor has drafted this decision to help you prepare a revised submission.Thank you for the responsive revisions. I would ask that you consider reviewer 1's concerns and add text to address these issues, particularly the issues of repeated use of the paradigm over multiple days and a comparison to previous head fixed olfaction methods. Reviewer 1 also suggests that adding in a platform condition with a back puff would make the control more complete. Please address this with either more experiments or a brief reason why more experiments would not enlighten.Reviewer #1:In their revision, Fink et al. present a substantially improved manuscript through the addition of data demonstrating that the burrow itself is actually needed for the VBA (Figure 2D-F). The addition of silicon probe single unit recording data during sharp movements is also a major improvement (Figure 7). These improvements address my previous main concerns. After reading the new manuscript, I have a few more concerns that I feel should be addressed before publication:1) The authors demonstrate that no training is required for the VBA, however, in order for the apparatus to be of maximal use in the community, the behaviors probed with the VBA should persist for many sessions/days. Figure 5 begins to get at this question (same mice over 2 different days), but I feel that this point should be addressed more directly; this could be accomplished with some added discussion, or with data that the authors may already have showing other VBA behavioral responses over days with the same mice.

The manuscript now includes a new figure (Figure 1—figure supplement 2) that describes the behavior of a cohort of mice tested in the VBA on four separate days over a 16-day interval. We find that repeated introduction into the VBA does not affect the animals’ behavior.

2) Figure 2E, F: for completeness, the puff back condition should be included for the platform case.

The puff back condition is now included for the platform case in Figure 2D-F.

3) Introduction, second paragraph: Since one major application of the VBA appears to be to olfactory responses (Figures 4 and 5), it seems appropriate to cite and discuss some previous methods using olfactory stimuli in head-fixed mice here: Lovett-Barron, 2014; Radvansky, 2018; Resulaj, 2015.

Radvansky, 2018, is now cited in the Introduction as an example of an olfactory behavioral paradigm that requires acclimation to head fixation and training, in contrast with the VBA.

We note that Lovett-Baron, 2014, was already given more detailed treatment in the Discussion regarding Figure 5 (with respect to the low temporal resolution of conditioned lick suppression) so we feel it is unnecessary to also cite it in the Introduction--especially since the focus of the second paragraph is on the challenges imposed by reliance on training-based paradigms, not on the temporal resolution of the assay. Since Guo, 2014, describes training animals to discriminate both somatosensory and olfactory stimuli, it seems redundant to also include a second reference (Resulaj, 2015) about olfaction.